# LoRA-Ensemble: Efficient Uncertainty Modelling for Self-Attention Networks

## Abstract

Numerous real-world decisions rely on machine learning algorithms and require calibrated uncertainty estimates. However, modern methods often yield overconfident, uncalibrated predictions. The dominant approach to quantifying the uncertainty inherent in the model is to train an ensemble of separate predictors and measure their empirical variance. In an explicit implementation, the ensemble has high computational cost and memory footprint, especially if the base model itself is already large, like modern transformers. This motivates efforts to develop implicit ensemble methods that emulate the ensemble without explicitly instantiating all its members. We introduce LoRA-Ensemble, a parameter-efficient ensembling method for self-attention networks. It is based on Low-Rank Adaptation (LoRA), originally developed for efficient LLM fine-tuning, and extends it into an implicit ensembling scheme, where all ensemble members share the same, pre-trained self-attention network, but have individual low-rank matrices for the attention projections. The resulting method not only outperforms state-of-the-art implicit techniques like BatchEnsemble, but even matches or exceeds the accuracy of an Explicit Ensemble, while at the same time achieving superior calibration.

## 1 Introduction

Machine learning models are increasingly applied also in fields where incorrect estimates can have severe consequences, e.g., autonomous driving, medical diagnosis, (extreme) weather event prediction, or decision support for agriculture. In such applications well-calibrated predictive uncertainties are crucial to enable self-diagnosis. Uncertainty can be separated into two components. *Aleatoric uncertainty*, a.k.a. irreducible noise, is inherent in the data. In contrast, *epistemic uncertainty* stems from a lack of knowledge about certain regions of the input space, due to a lack of training data Der Kiureghian & Ditlevsen (2009).

Quantification of epistemic uncertainty in large machine learning models is non-trivial. Analytical computation is usually intractable, thus research has focused on efficient approximations Graves (2011); Blundell et al. (2015); Welling et al. (2011). To date, probabilistic ensembles remain the best-performing approach Lakshminarayanan et al. (2017). In a naïve implementation, such an ensemble consists of multiple independently trained models. Individual models are interpreted as Monte Carlo samples from the posterior weight space and are used to obtain an unbiased estimator of the posterior distribution. To achieve a low correlation between ensemble members one can capitalize on the stochastic nature of the training process and start from different initial weights, and/or sample different random batches of data. The basic principle is that the predictions of different ensemble members will agree near observed training samples, whereas they may vary far away from the training data. Their spread therefore serves as a measure of epistemic uncertainty. Empirically, even small ensembles often capture the uncertainty well (in expectation), i.e., they are well calibrated.

An issue with naïve ensembles is that their computational cost and memory footprint grow proportional to the number of ensemble members. For smaller models, the added cost and energy use may be acceptable. But for modern neural networks with up to several billion parameters, hardware restrictions render the naïve approach intractable, in particular, one can no longer hold the entire ensemble in memory. Consequently, research has focussed on ways to create ensembles implicitly, without requiring multiple copies of the full base model Wen et al. (2020); Wenzel et al. (2020);

Huang et al. (2017); Turkoglu et al. (2022). Unfortunately, most of these parameter-efficient ensembling techniques are not applicable to the newest generation of neural networks. Transformer networks Vaswani et al. (2017) have become popular due to their superior ability to capture complex structures in data. However, implicit ensembling schemes tend to underperform for transformers, as demonstrated in our experiments, or are incompatible with them, as detailed in Appendix V.

Several studies have shown that modern neural networks are heavily overparametrized and that their results have low intrinsic dimension Li et al. (2018a); Aghajanyan et al. (2020). This led Hu et al. (2021) to propose Low-Rank Adaptations (LoRAs) as a way of fine-tuning Large Language Models (LLMs) for different tasks while avoiding the prohibitively large memory and compute requirements of retraining them. It turns out that the weight matrices in such models can be factorized to have very low rank, with hardly any loss in prediction performance.

We show that LoRA can also serve as a basis for a novel, parameter-efficient ensemble method tailored to the transformer architecture. In line with the trend towards parameter-efficient fine-tuning, our method uses a pre-trained transformer model, which is expanded into an implicit ensemble by varying the LoRA factorization, while keeping the backbone weights frozen. In this way, our method requires a small number of additional parameters to turn an existing transformer model into a diverse ensemble whose performance across various tasks is comparable to an Explicit Ensemble. In summary, our contributions are:

- We introduce LoRA-Ensemble, a parameter-efficient probabilistic ensemble method for self-attention networks.
- LoRA-Ensemble can be readily combined with most pre-trained transformer networks, irrespective of their specific architecture and application domain: it simply replaces the linear projection layers in the attention module with LoRA layers.
- We apply LoRA-Ensemble to different classification tasks, including conventional image labeling, skin lesions classification in dermatoscopic images, fine-grained image classification, sound classification, out-of-distribution (OOD) detection, and language modeling; and demonstrate significant gains in accuracy and uncertainty modeling.
- We demonstrate LoRA-Ensemble outperforms traditional Explicit Ensembles by fostering greater diversity among members, both in their learned functions and in weight space.
- We conduct extensive empirical analyses of how LoRA rank, initialization scheme, model scale, and parameter-sharing strategies impact performance, and we adapt LoRA-Ensemble for convolutional neural networks (CNNs) to demonstrate its broad applicability.

## 2 LoRA-Ensemble

The Low-Rank Adaptation (LoRA) technique makes it possible to use a pre-trained model and fine-tune it without having to retrain all its parameters. This is particularly beneficial for modern neural networks with large parameter spaces. The underlying principle is to freeze the pre-trained model weights $W_0 \in \mathbb{R}^{k \times d}$ and instead constrain the updates to a low-rank decomposition. This can be expressed mathematically as:

$$W = W_0 + \Delta W = W_0 + B \cdot A . \tag{1}$$

Here $B \in \mathbb{R}^{k \times r}$ and $A \in \mathbb{R}^{r \times d}$ are two trainable low-rank matrices, where $r \ll \min(d, k)$. $W$ and $\Delta W$ are then multiplied with the same input $x$, which yields the following modified forward pass:

$$h = W_0 \cdot x + \Delta W \cdot x = W_0 \cdot x + B \cdot A \cdot x . \tag{2}$$

LoRA applies its low-rank adaptation scheme exclusively to the weight matrices of the self-attention modules in a transformer, while leaving the interleaved MLP layers untouched. Concretely, the adapted weights are $W_q$, $W_k$, and $W_v$, which project the input into query, key, and value representations, along with $W_o$, which merges the outputs of the attention heads. As in Hu et al. (2021), the projection matrices are treated as single units, disregarding their typical partitioning into multiple attention heads. In Appendix G, we provide additional ablations on the placement of LoRA layers within the transformer, as well as on ensemble design choices, illustrating how these factors impact predictive performance, calibration, and efficiency.

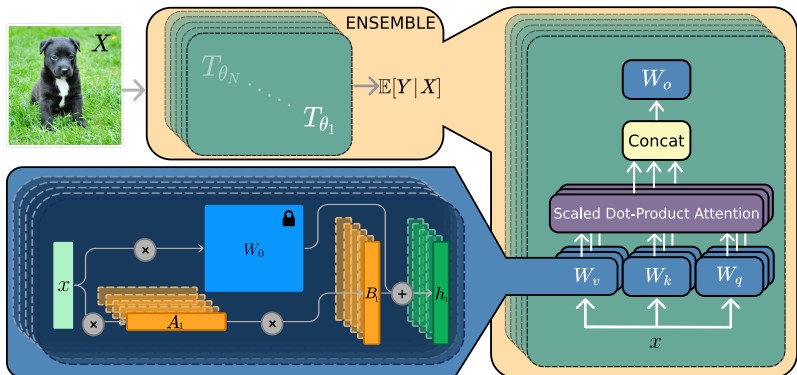

Figure 1: A schema of a LoRA-Ensemble. The computation structure of the multi-head self-attention module (right), and LoRA-Ensemble module (bottom left). $X$ denotes the actual input, and $x$ represents the intermediate input representation.

Although not designed with uncertainty calibration in mind, the LoRA concept fulfills all the requirements of an implicit deep ensemble: By modifying the weights of the highly nonlinear self-attention mechanism one is able to generate a diverse collection of networks with the same architecture and objective. By learning an additive, low-rank update $\Delta W = B \cdot A$ rather than directly tuning the weight matrices, the expansion into a model ensemble adds only a small number of parameters and is efficient. In detail, we start from a single, pre-trained model with frozen parameters $W_0$ and expand it with a set of trainable low-rank matrices $\Delta W_i, \forall i = 1 \dots N$. At each transformer block, there now is a different forward pass per ensemble member $i$, as illustrated in Fig. 1:

$$h_i = W_0 \cdot x + \Delta W_i \cdot x = W_0 \cdot x + B_i \cdot A_i \cdot x \; , \tag{3}$$

leading to $N$ different predictions $T_{\theta_i}(X)$ for a given input $X$. From those individual predictions, we compute the ensemble mean and variance in the standard manner:

$$\mathbb{E}[Y|X] \approx \frac{1}{N} \sum_{i=1}^{N} T_{\theta_i}(X) \quad , \quad \mathrm{Var}[Y|X] \approx \frac{1}{N} \sum_{i=1}^{N} \left( T_{\theta_i}(X) - \mathbb{E}[Y|X] \right)^2 \; . \tag{4}$$

Refer to Appendices L, M, and N, respectively, for implementation, training, and initialization details. We publicly release the PyTorch implementation of LoRA-Ensemble, along with pre-trained weights, on GitHub.

## 3 EXPERIMENTS

In the following section, we evaluate the proposed LoRA-Ensemble on several datasets with regard to its predictive accuracy, uncertainty calibration, and memory usage. For each experiment we also show $1 \cdot \sigma$ error bars, estimated from five independent runs with different random initializations.

As a first sandbox experiment, we perform image classification for the popular, widely used CIFAR-100 benchmark Krizhevsky (2009) (see Appendix A for CIFAR-10 experiments). The dataset consists of 100 object classes, each with 600 samples, for a total size of 60 000 images. From that set, 10 000 images are designated as test data, with all classes equally distributed between the training and testing portions.

The HAM10000 dataset was proposed for the *Human Against Machine with 10 000 training images* study Tschandl et al. (2018). It consists of 10 015 dermatoscopic images of pigmented skin lesions, collected from different populations. The dataset was initially assembled to compare machine learning methods against medical professionals on the task of classifying common pigmented skin lesions. Compared to CIFAR-100, this is arguably a more relevant test bed for our method: in the medical domain, uncertainty calibration is critical, due to the potentially far-reaching consequences of incorrect diagnoses and treatment planning.

For both datasets, LoRA-Ensemble is evaluated against a range of baselines. As a sanity check, we include a single Vision Transformer (ViT) model as well as a ViT model augmented with LoRA in

the attention modules. These models lack dedicated mechanisms for uncertainty calibration, instead relying on class-conditional likelihoods to quantify uncertainty. We further compare against an Explicit Ensemble and several common implicit approaches: (i) Monte Carlo Dropout (MC Dropout) as implemented in Li et al. (2023), (ii) Batch-Ensemble Wen et al. (2020), and (iii) Snapshot Ensemble Huang et al. (2017), with implementation details provided in Appendix I and U. For challenges related to other implicit methods, we refer to Appendix V. In addition, we compare with L2 (Kim et al., 2021) and LRFormer (Ye et al., 2023), two recent methods for uncertainty modeling in ViTs, in our HAM10000 experiments, as well as with the recently proposed Bayesian-LoRA (Yang et al., 2024), which improves uncertainty calibration in fine-tuned LLMs, in our SST-2 experiments. The LoRA rank was empirically set to 8 for CIFAR-100 and 4 for HAM10000.

To demonstrate that LoRA-Ensemble scales to large, fine-grained, real-world datasets, we apply it to iNaturalist 2017 Van Horn et al. (2018), comprising 675 170 images across 5 089 species, an order of magnitude larger than CIFAR-100. Severe class imbalance, high intra-class variability, and subtle inter-class differences make uncertainty quantification especially challenging, due to the difficulty to avoid overconfident errors among similar species, and to flag uncertain predictions for rare species. The LoRA rank was set to 64 for this experiment. Refer to Appendix C for detailed sensitivity analysis of the LoRA rank, together with a practical guide for hyperparameter selection.

As a further benchmark from a different application domain, we process the ESC-50 environmental sounds dataset Piczak (2015). It consists of 2 000 sound samples, each five seconds long, that represent 50 different semantic classes with 40 samples each. To prepare the raw input waveforms for analysis, they are converted into 2-dimensional time/frequency spectrograms, see Gong et al. (2021). These spectrograms form the input for Audio Spectrogram Transformer, a state-of-the-art transformer model for sound classification. We also extend our evaluation to natural language processing with the SST-2 sentiment classification dataset, using BERT base uncased Socher et al. (2013); Devlin et al. (2019). Results for both datasets are provided in Appendix A.2 and Appendix A.7.

We evaluated each method's predictive performance using classification accuracy and F1 score, and its calibration quality through Expected Calibration Error (ECE), Negative Log-Likelihood (NLL), and Brier score. The ECE measures how far predicted confidences deviate from observed error rates, i.e., perfect calibration occurs when the estimated uncertainties match the actual likelihood of a mis-classification. The definitions of all metrics are given in Appendix X.

For the out-of-distribution (OOD) experiment, we trained models on CIFAR-100 and evaluated them using in-distribution samples from CIFAR-100 and OOD samples from CIFAR-10 or SVHN Netzer et al. (2011), following standard practice Hendrycks & Gimpel (2016). Performance was measured using AUROC and AUPRC.

**Compute Cost.**   LoRA-Ensemble is markedly lighter than Explicit Ensembles, requiring far fewer parameters and memory (about 14x cheaper for 16 members), while also training about 2x faster and delivering over 3x faster inference (Tab. 9; Appendix A.9). Timings are based on our PyTorch vmap implementation, which introduces a one-time overhead and is not fully optimized (see Appendix L).

**CIFAR-100.**   Quantitative results are summarized in Tab. 1. Reliability diagrams, along with plots depicting classification accuracy and ECE as a function of ensemble size, are provided in Appendix A.3.

LoRA-Ensemble consistently reaches higher accuracy than MC Dropout and Snapshot Ensemble, with a notable edge of approximately 5 percentage points. Despite its conceptual similarity to the LoRA-Ensemble, the Batch-Ensemble is the weakest performer among all methods when applied to transformers. Appendix I examines this finding in detail and outlines key distinctions between the two approaches. Surprisingly, LoRA-Ensemble also consistently surpasses the Explicit Ensemble by about 2 percentage points, apparently a consequence of the fact that already a single ViT model, and thus every ensemble member, benefits from the addition of LoRA.

The LoRA-Ensemble also achieves better-calibrated predictive uncertainties than all implicit ensembling methods and the Explicit Ensemble. Interestingly, although a single LoRA network is already very well calibrated, forming an ensemble slightly degrades its calibration, an effect not observed for the NLL or Brier score (Tab. 1). The reliability diagram in Fig. 5 in the appendix somewhat elucidates this unexpected behavior: LoRA-Ensemble is under-confident on CIFAR-100, i.e., its

Table 1: Model performance on the CIFAR-100 dataset for the compared methods. Ensembles have 16 members. Best score for each metric in **bold**, second-best underlined.

| Method | Accuracy (↑) | F1 (↑) | ECE (↓) | NLL (↓) | Brier (↓) |
|---|---|---|---|---|---|
| Single Network | 76.6 ± 0.3 | 76.6 ± 0.3 | 0.145 ± 0.004 | 1.181 ± 0.019 | 0.370 ± 0.004 |
| Single Net w/ LoRA | 79.6 ± 0.2 | 79.4 ± 0.2 | **0.014** ± 0.003 | 0.671 ± 0.005 | 0.286 ± 0.003 |
| MC Dropout | 77.1 ± 0.5 | 77.2 ± 0.4 | 0.055 ± 0.002 | 1.138 ± 0.014 | 0.336 ± 0.005 |
| Snapshot Ensemble | 77.0 ± 0.1 | 77.2 ± 0.2 | 0.123 ± 0.002 | 4.416 ± 0.046 | 1.614 ± 0.007 |
| Batch-Ensemble | 68.8 ± 0.1 | 68.5 ± 0.1 | 0.102 ± 0.002 | 1.093 ± 0.002 | 0.437 ± 0.001 |
| Explicit Ensemble | 79.8 ± 0.1 | 79.8 ± 0.2 | 0.100 ± 0.001 | 0.745 ± 0.003 | 0.284 ± 0.002 |
| LoRA-Ensemble | **82.5** ± 0.1 | **82.5** ± 0.1 | 0.035 ± 0.001 | **0.587** ± 0.001 | **0.253** ± 0.000 |

predictions are more accurate than its confidence suggests. As noted by Rahaman & Thiery (2020), ensembling under-confident models can worsen calibration since accuracy grows faster than confidence. While under-confidence may be preferable in safety-critical settings, where over-estimating uncertainty is safer than being over-confident, we show in Appendix K that simple post-hoc Temperature Scaling effectively corrects this and yields near-perfect calibration.

Table 2: Model performance on the HAM10000 dataset for the compared methods. Ensembles have 16 members. Best score for each metric in **bold**, second-best underlined
.

| Method | Accuracy (↑) | F1 (↑) | ECE (↓) | NLL (↓) | Brier (↓) |
|---|---|---|---|---|---|
| Single Network | 84.1 ± 0.3 | 71.4 ± 0.7 | 0.139 ± 0.004 | 1.138 ± 0.040 | 0.291 ± 0.009 |
| Single Net w/ LoRA | 83.2 ± 0.7 | 70.7 ± 1.3 | 0.085 ± 0.004 | 0.569 ± 0.027 | 0.256 ± 0.011 |
| LRFormer | 74.3 ± 1.9 | 52.1 ± 3.2 | 0.053 ± 0.022 | 0.737 ± 0.014 | 0.354 ± 0.011 |
| L2 | 74.1 ± 1.8 | 50.7 ± 3.9 | 0.065 ± 0.024 | 0.766 ± 0.036 | 0.360 ± 0.021 |
| MC Dropout | 83.7 ± 0.4 | 71.0 ± 0.9 | 0.099 ± 0.007 | 0.631 ± 0.023 | 0.270 ± 0.009 |
| Snapshot Ensemble | 84.9 ± 0.3 | 73.7 ± 0.9 | 0.058 ± 0.004 | 0.431 ± 0.007 | 0.217 ± 0.004 |
| Batch-Ensemble | 76.8 ± 1.6 | 58.4 ± 2.8 | 0.064 ± 0.021 | 0.651 ± 0.003 | 0.332 ± 0.002 |
| Explicit Ensemble | 85.8 ± 0.2 | 74.6 ± 0.4 | 0.105 ± 0.002 | 0.536 ± 0.007 | 0.218 ± 0.002 |
| LoRA-Ensemble | **88.0** ± 0.2 | **78.3** ± 0.6 | **0.037** ± 0.002 | **0.342** ± 0.003 | **0.175** ± 0.002 |

**HAM10000 Lesion Classification.** In many medical applications, well-calibrated models are essential. As a test case, we use the classification of pigmented skin lesions and again compare the same group of models in terms of accuracy and calibration. The results are summarized in Tab. 2. Similar to the CIFAR-100 evaluation, LoRA-Ensemble outperforms all other methods by a clear margin, with respect to both classification accuracy and calibration.

The experiments also further support the above discussion of confidence vs. ensemble size (Sec. 3). For HAM10000, LoRA-Ensemble is slightly over-confident (just like the Explicit Ensemble) and, indeed, its calibration error decreases with ensemble size in this case, see Appendix A.4.

We conducted further experiments on HAM10000 using different backbone architectures (DeiT) with varying numbers of parameters. See Tab. 10 in Appendix B. LoRA-Ensemble generalizes smoothly across different backbones, and as the number of parameters in the backbone increases, its advantage over the Explicit Ensemble becomes more pronounced, in both accuracy and calibration. For generalization to the CNN architecture, see Appendix H.

**Large-Scale Fine-Grained Image Classification with iNaturalist.** On iNaturalist 2017 (INat2017), our LoRA-Ensemble almost matches the Explicit Ensemble in accuracy, while substantially improving the calibration, using only a fraction of the parameters and compute. This demonstrates that the method scales well and enables reliable uncertainty estimation for large, fine-grained, imbalanced datasets. Refer to Appendix F for additional results.

**Out-of-Distribution Detection & Dataset Shift Robustness.** To evaluate LoRA-Ensemble for OOD detection, a crucial aspect of handling uncertainty in deep learning models Hendrycks & Gimpel (2016), we run an experiment in which models were trained on CIFAR-100 (in-distribution) and

Table 3: Performance on the INat2017 dataset for all compared methods using three different random seeds. Ensembles have 4 members. Best score for each metric in **bold**, second-best underlined.

| Method | Accuracy ($\uparrow$) | F1 ($\uparrow$) | ECE ($\downarrow$) | NLL ($\downarrow$) | Brier ($\downarrow$) |
|---|---|---|---|---|---|
| Single Network | $42.6 \pm 0.2$ | $37.8 \pm 0.2$ | $0.293 \pm 0.002$ | $1.054 \pm 0.001$ | $0.207 \pm 0.001$ |
| Single Net w/ LoRA | $47.7 \pm 0.1$ | $43.1 \pm 0.1$ | $\underline{0.096} \pm 0.001$ | $\underline{0.662} \pm 0.001$ | $0.166 \pm 0.000$ |
| MC Dropout | $47.5 \pm 0.1$ | $40.3 \pm 0.1$ | $0.206 \pm 0.002$ | $0.895 \pm 0.002$ | $0.172 \pm 0.000$ |
| Explicit Ensemble | $\mathbf{49.6} \pm 0.2$ | $\mathbf{44.6} \pm 0.3$ | $0.199 \pm 0.002$ | $0.716 \pm 0.002$ | $\underline{0.165} \pm 0.000$ |
| LoRA-Ensemble | $\underline{49.3} \pm 0.1$ | $\underline{44.1} \pm 0.2$ | $\mathbf{0.045} \pm 0.001$ | $\mathbf{0.610} \pm 0.000$ | $\mathbf{0.160} \pm 0.000$ |

Table 4: Model performance on the OOD task. CIFAR-100 is used as the in-distribution dataset and CIFAR-10 and SVHN as the out-of-distribution dataset. Ensembles for all methods consist of 16 members. Results for Split-Ensemble are taken from Chen et al. (2024). The best score for each metric is highlighted in **bold**, with the second-best score underlined.

| OOD Dataset | CIFAR-10 | | SVHN | |
|---|---|---|---|---|
| Method | AUROC ($\uparrow$) | AUPRC ($\uparrow$) | AUROC ($\uparrow$) | AUPRC ($\uparrow$) |
| Split-Ensemble Chen et al. (2024) | 79.2 | 81.7 | 81.2 | 69.9 |
| Single Network | $75.6 \pm 0.3$ | $77.6 \pm 0.6$ | $76.4 \pm 1.8$ | $67.1 \pm 2.3$ |
| Single Network with LoRA | $\underline{80.1} \pm 0.5$ | $\underline{82.4} \pm 0.6$ | $\underline{85.9} \pm 0.9$ | $\underline{75.4} \pm 1.7$ |
| MC Dropout | $75.1 \pm 0.5$ | $73.7 \pm 0.9$ | $52.3 \pm 12.4$ | $29.9 \pm 7.1$ |
| Explicit Ensemble | $78.9 \pm 0.2$ | $80.8 \pm 0.2$ | $74.8 \pm 1.3$ | $63.9 \pm 1.5$ |
| LoRA-Ensemble | $\mathbf{82.1} \pm 0.1$ | $\mathbf{84.1} \pm 0.1$ | $\mathbf{89.9} \pm 0.6$ | $\mathbf{80.9} \pm 1.0$ |

tested on samples from both CIFAR-100 and CIFAR-10 or SVHN (out-of-distribution). Following Sim et al. (2023) and Chen et al. (2024), we use the maximum softmax probability as the confidence score. Table 4 highlights that LoRA-Ensemble achieves superior performance compared to all other methods across both settings and metrics, surpassing even the recently proposed Split-Ensemble approach Chen et al. (2024) that was specifically designed for OOD tasks. Furthermore, consistent with our earlier observations, even a single LoRA model outperforms the Explicit Ensemble, highlighting its robustness in OOD scenarios. To further assess robustness under distribution shifts, we also experimented with the CIFAR-10/100-C benchmarks across various severity levels. As detailed in Appendix A.8, LoRA-Ensemble consistently achieves superior accuracy and calibration compared to all baselines, maintaining reliable uncertainty estimates even under severe corruptions.

## 4 ENHANCED DIVERSITY IN LoRA-ENSEMBLE

To better understand the behavior of LoRA-Ensemble, we explore the diversity of its members and compare it to the Explicit Ensemble. The experiments are run on HAM10000 with 16 ensemble members. Diversity is crucial for effective ensembles, as highly correlated members offer little added value Zhang (2012). If an ensemble contains diverse parameter configurations that equally explain observations, then it will more comprehensively capture the epistemic uncertainty Kendall & Gal (2017). For empirical evidence, refer to Appendix E.

Following Fort et al. (2019b), we first assess function space diversity through the predictions of individual ensemble members. In Fig. 2, we first compute the disagreement rate on the test set, defined as $\frac{1}{N} \sum_{n=1}^{N} \mathbb{I}[T_{\theta_i}(X_n) \neq T_{\theta_j}(X_n)]$, where $T_{\theta_i}(X_n)$ represents the class label predicted by ensemble member $i$ for input $X_n$, and $\mathbb{I}$ is the indicator function. Next, we construct a probability distribution for each ensemble member by aggregating their softmax outputs across all test samples, then compute pairwise Jensen-Shannon divergences (JSD). Finally, we use t-SNE Van der Maaten & Hinton (2008) to visualize their spread in function space (aggregated softmaxes). The analysis reveals that LoRA-Ensemble exhibits significantly higher diversity among ensemble members compared to an Explicit Ensemble. I.e., LoRA-Ensemble appears to capture a wider range of modes in function space.

We further inspect the weight spaces of LoRA-Ensemble and Explicit Ensemble with spectral analysis, focusing on the projection matrices within the attention blocks of the ViT (Base-32) model

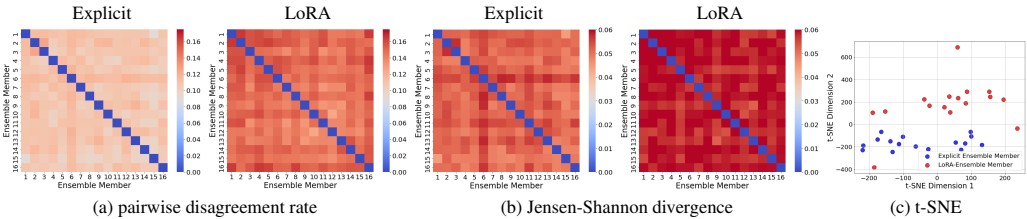

Figure 2: Function space analysis of LoRA-Ensemble vs. Explicit Ensemble.

pre-trained on ImageNet. We show the analysis for value projection matrices, given their strong association with learned representations; details for query and key projection matrices can be found in Appendix D. We employ Singular Value Decomposition (SVD) to identify the most significant transformations encoded in the weights, following the logic that larger singular values correspond to the most impactful components. As proposed by Shuttleworth et al. (2024), we analyze the similarity between the initial (pre-trained) weights and the final trained weights of ensemble members. LoRA-Ensemble and Explicit Ensemble lead to very different parameter updates. LoRA-Ensemble introduces new high-ranking singular vectors that are near-orthogonal to those in the initial weights, referred to as "intruder dimensions" Shuttleworth et al. (2024). In contrast, Explicit Ensemble members tend to adhere closely to the spectral structure of the initial weights (see Fig. 13 in Appendix).

The random initialization of matrices $A$ and $B$ in the LoRA module leads to an intriguing phenomenon: the intruder dimensions of different LoRA-Ensemble members are near-orthogonal, as shown by the cosine similarities between the highest-ranking singular vectors of different members in Fig. 3 (for details see Appendix D). The figure shows similarity values averaged over layers and pairs of members, for rank 4. Notably, the highest-ranked singular vectors of distinct members exhibit almost no similarity; in contrast to the Explicit Ensemble, where they are highly correlated. The weight-space cosine similarity provides further evidence of enhanced diversity. LoRA-Ensemble members exhibit greatly increased diversity in weight space. To visualize training trajectories, we apply t-SNE to plot the evolution of the model weights during training. LoRA-Ensemble members span a larger part of the loss landscape, indicating diverse learning dynamics. In contrast, Explicit Ensemble members remain closer to the initial weights, reflecting reduced diversity. Overall, these results suggest LoRA-Ensemble better explores the weight space, and thus the epistemic uncertainty.

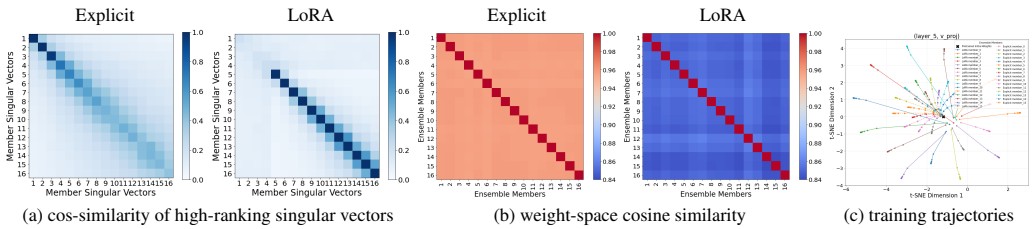

Figure 3: Weight space analysis of LoRA-Ensemble vs. Explicit Ensemble.

## 5 DISCUSSION

**Effectiveness of LoRA-Ensemble.** Across diverse tasks, our experiments consistently show that LoRA-Ensemble matches or surpasses the predictive performance of the state-of-the-art Explicit Ensemble while offering superior calibration. Adding LoRA to a single model *without any ensembling* improves calibration in most experiments beyond that of a 16-member Explicit Ensemble. This effect may be linked to the well-documented over-parameterization of modern neural networks, which often achieve higher predictive accuracy at the cost of poorer calibration (e.g., Guo et al., 2017). By incorporating LoRA while treating all pre-trained weights as constants, we significantly reduce the trainable parameter space, potentially favoring better calibration. However, limiting trainable parameters alone does not ensure better accuracy or calibration, e.g., many forms of regularization or selective training may fall short. We believe that the effectiveness of LoRA-Ensemble stems from its unique learning dynamics, which we explore in Sec. 4 and Appendix D. Its members converge

to diverse solutions that span a broader area of the loss landscape, enabling better exploration of the weight space and more effective estimates of epistemic uncertainty. Increasing the number of members in the LoRA-Ensemble enhances predictive power, potentially improving accuracy while maintaining good calibration due to the limited number of trainable weights. However, if the trainable weights are not limited, e.g. when increasing the LoRA rank too far, calibration can worsen, as shown in Fig.11a, Tab.7. This effect aligns with the findings of Shuttleworth et al. (2024), which indicate that excessively increasing the rank of LoRA may cause it to lose its unique learning dynamics. Furthermore, jointly training the backbone and optimizing all parameters simultaneously degrades performance, see Appendix F for details. Conversely, enhancing predictive power by increasing the pre-trained weights (while keeping trainable weights constant) further improves the effectiveness of the LoRA-Ensemble, see Appendix B. Lastly, the LoRA-Ensemble remains effective even when pre-training is not available; see Appendix J for the experiment where it was pre-trained and fine-tuned on the same target dataset.

**Comparison to Bayesian LoRA.** On the language modeling task (SST-2 sentiment classification), LoRA-Ensemble consistently outperforms Bayes-LoRA Yang et al. (2024) across accuracy, F1, NLL, and Brier score, with Bayes-LoRA only achieving a marginally lower ECE. Refer to Tab. 8 and Appendix A.7. This result is consistent with previous works, showing that Laplace-based methods improve calibration at the expense of predictive accuracy Deng et al. (2022) or often fall short compared to ensembles Daxberger et al. (2021); Eschenhagen et al. (2021). Notably, in our experiments, LoRA-Ensemble is more than 10x faster at inference, demonstrating that it combines strong accuracy, reliable calibration, and efficiency. In contrast, the Laplace-based Bayesian method trades accuracy for improved uncertainty estimates while remaining less efficient. Refer to Appendix W for more details and discussion.

**Limitations & Future Work.** Despite its memory-efficient design and reduced per-member training and inference overhead, our ensembling approach maintains computational complexity similar to that of conventional ensembles, since each batch still requires separate forward passes for every member. As discussed by Rahaman & Thiery (2020), our work also suggests that in a high-parameter regime, deep ensembles may not exhibit the same behavior as they do in a low-parameter regime, where they typically improve calibration properties. This type of phase shift in the bias-variance trade-off, the so-called Double Descent Phenomenon, has previously been observed for large neural networks Nakkiran et al. (2021). It would be valuable to conduct an in-depth analysis of how deep ensembles behave in high-parameter regimes.

## 6 RELATED WORK

**Estimation of Epistemic Uncertainty.** A lot of work has gone into estimating the epistemic uncertainty in Artificial Neural Network (ANN). As the analytical computation of the posterior in such models is generally intractable, methods for approximate Bayesian inference have been proposed. Such methods rely on imposing an appropriate prior on the weights and using the likelihood of the training data to get an approximate posterior of the weight space.

The main techniques are, on the one hand, Variational Inference Graves (2011); Ranganath et al. (2014), which Blundell et al. (2015) have specialized to neural networks as *Bayes by Backprop*. And on the other hand variants of Markov Chain Monte Carlo (MCMC) Neal (1996); Chen et al. (2014), including Stochastic Gradient Langevin Dynamics (SGLD) Welling et al. (2011). These, however, are often not able to accurately capture high-dimensional and highly non-convex loss landscapes, like the ones usually encountered in deep learning Gustafsson et al. (2019). More recently, Bayesian LoRA methods have been explored, with Yang et al. (2024) using a Laplace approximation for improved calibration and Wang et al. (2024) jointly learning mean and covariance during fine-tuning.

**Ensembles and Implicit Ensembling.** Lakshminarayanan et al. (2017) have proposed a method known as deep ensembles. It uses a set of neural networks with identical architecture that are independently and randomly initialized, and (as usual) trained with variants of Stochastic Gradient Descent (SGD). While the latter introduces further stochasticity, Fort et al. (2019a) have shown that the initialization of the weights is more important to explore the admissible weight space. Ensemble members will generally converge to different modes of the loss function, such that they can be considered Monte Carlo samples of the posterior distribution Wilson & Izmailov (2020); Izmailov

et al. (2021). While ensembles, in general, yield the best results in terms of accuracy and uncertainty calibration, a straightforward implementation suffers from high memory and compute requirements, since multiple instances of the full neural network must be trained and stored. This can become prohibitive for modern neural networks with many millions, or even billions, of parameters.

Consequently, researchers have attempted to find ways of mimicking the principle of deep ensembles without creating several full copies of the base model. Gal & Ghahramani (2015) have proposed Monte Carlo Dropout, where the posterior is approximated by sampling different dropout patterns at inference time. While this is less expensive in terms of memory, performance is often worse. Masksembles Durasov et al. (2020) are a variant that attempts to select suitable dropout masks in order to obtain better uncertainty estimates. Snapshot Ensembles Huang et al. (2017) use cyclic learning rates to steer the learning process such that it passes through multiple local minima, which are then stored as ensemble members. This reduces the training effort but does not address memory requirements or inference time.

Particularly relevant for our work are attempts that employ a shared backbone and modify only selected layers. Havasi et al. (2020) follow that strategy, in their case only the first and last layer of a neural network are replicated and trained independently to emulate an ensemble. Packed-Ensemble Laurent et al. (2023) leverage grouped convolutions to train lightweight ensembles within a single shared backbone. Batch-Ensemble Wen et al. (2020) is similar to LoRA-Ensemble in that it also uses low-rank matrices to change the model parameters. More specifically, shared weight matrices are modulated by element-wise multiplication with different rank-1 matrices to achieve the behavior of a deep ensemble while adding only a small number of parameters. Wenzel et al. (2020) take this concept further by also ensembling over different hyper-parameter settings. Turkoglu et al. (2022) freeze all weights of the base model and instead vary the feature-wise linear modulation (FiLM, Li et al., 2018b; Takeda et al., 2021). A related concept was recently introduced for LLMs: the Mixtral of Experts model Jiang et al. (2024) averages over a sparse mixture of experts to efficiently generate text.

**Low-Rank Adaptation in Transformer Networks.** Low-Rank Adaptation was originally conceived as a parameter-efficient way of fine-tuning Large Language Models (LLMs) Hu et al. (2021). It is based on the observation that, while modern neural networks have huge parameter spaces, the solutions they converge to have much lower intrinsic dimension Li et al. (2018b); Aghajanyan et al. (2020). LoRA exploits this and Hu et al. (2021) show that even when fine-tuning only low-rank update matrix $B \cdot A$ (sometimes with rank as low as one or two), the resulting models are competitive with much more expensive fine-tuning schemes. The method quickly became popular and has since also been extended with weight-decomposition Liu et al. (2024). The Low-Rank Adaptation (LoRA) idea has been applied in various fields, notably for denoising diffusion models Luo et al. (2023); Golnari (2023). As we have shown, the LoRA adaptation mechanism naturally lends itself to parameter-efficient ensembling, which we investigate in the context of uncertainty calibration, with a primary focus on vision transformers but not limited to them. A similar idea has concurrently been explored for fine-tuning LLMs Wang et al. (2023), yielding promising results in both predictive performance and uncertainty estimation.

# 7 CONCLUSION

We have presented LoRA-Ensemble, a novel, parameter-efficient method for probabilistic learning that is tailored to the transformer architecture (and potentially other architectures that make use of the attention mechanism). LoRA-Ensemble uses a simple, but efficient trick to turn a single base model into an implicit ensemble: the weights of the base model are kept frozen, but are modulated with the Low-Rank Adaptation mechanism. By training multiple, stochastically varying instances of the low-rank matrices that define the modulation, one obtains a diverse set of ensemble members that share the majority of their weights and introduce only minimal overhead through the coefficients of their individual low-rank matrices. Our extensive experiments demonstrate that the proposed approach excels in both predictive performance and uncertainty calibration. Not only does it surpass other state-of-the-art implicit ensembling methods, but it also outperforms Explicit Ensembles on many tasks. This challenges the prevailing notion in the literature that Explicit Ensembles represent the upper bound for efficient ensembling methods (Wen et al., 2020).

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

# APPENDIX

## CONTENTS

## A  ADDITIONAL EXPERIMENTS AND RESULTS

This section presents comprehensive experimental results for the newly introduced CIFAR-10 dataset, Environmental Sound Classification on ESC-50. It also includes additional results for the CIFAR-100 and HAM10000 datasets and expanded baseline comparisons for CIFAR-100.

### A.1  CIFAR-10

The results for the CIFAR-10 dataset, as shown in Tab. 5, indicate that LoRA-Ensemble outperforms all other methods. Close behind is a single network enhanced with LoRA. This mirrors the results found in the main paper for CIFAR-100, with the exception of the calibration for a single model. It is important to note that although all methods achieve high accuracy and the differences between them are minimal, calibration is nearly perfect for most approaches. This suggests that the CIFAR-10 dataset is relatively easy for modern transformer models, and the results should not be over-interpreted. Nevertheless, the consistent performance across different random seeds suggests that the ranking is likely significant. Given the balanced nature of the CIFAR-10 dataset, the accuracy and F1-score are almost identical.

Table 5: Performance on the CIFAR-10 dataset for all compared methods. Ensembles have 16 members. Best score for each metric in **bold**, second-best underlined.

| Method | Accuracy ($\uparrow$) | F1 ($\uparrow$) | ECE ($\downarrow$) | NLL ($\downarrow$) | Brier ($\downarrow$) |
|---|---|---|---|---|---|
| Single Network | $92.8 \pm 0.1$ | $92.8 \pm 0.1$ | $0.051 \pm 0.001$ | $0.333 \pm 0.003$ | $0.120 \pm 0.002$ |
| Single Net w/ LoRA | $\underline{94.5} \pm 0.0$ | $\underline{94.5} \pm 0.0$ | $\underline{0.009} \pm 0.001$ | $\underline{0.163} \pm 0.002$ | $\underline{0.082} \pm 0.001$ |
| MC Dropout | $92.9 \pm 0.2$ | $92.9 \pm 0.2$ | $0.023 \pm 0.002$ | $0.260 \pm 0.005$ | $0.110 \pm 0.003$ |
| Snapshot Ensemble | $93.1 \pm 0.1$ | $93.1 \pm 0.1$ | $0.037 \pm 0.002$ | $1.062 \pm 0.021$ | $0.510 \pm 0.008$ |
| Batch-Ensemble | $88.5 \pm 0.1$ | $88.5 \pm 0.1$ | $0.048 \pm 0.001$ | $0.347 \pm 0.001$ | $0.172 \pm 0.000$ |
| Explicit Ensemble | $94.1 \pm 0.1$ | $94.1 \pm 0.1$ | $0.031 \pm 0.001$ | $0.181 \pm 0.002$ | $0.087 \pm 0.001$ |
| LoRA-Ensemble | $\mathbf{95.9} \pm 0.1$ | $\mathbf{95.9} \pm 0.1$ | $\mathbf{0.003} \pm 0.001$ | $\mathbf{0.128} \pm 0.001$ | $\mathbf{0.064} \pm 0.000$ |

### A.2  ESC-50 ENVIRONMENTAL SOUND CLASSIFICATION

Like for the ViT model, we train an Audio Spectrogram Transformer version of LoRA-Ensemble by modifying the attention layers with different sets of LoRA weights. That ensemble is then compared to a single instance of AST with and without LoRA, to an Explicit Ensemble of AST-models, and to an MC Dropout variant of AST, similar to Li et al. (2023). For ESC-50 a LoRA rank of 16 worked best, presumably due to the larger domain gap between (image-based) pre-training and the actual audio classification task. The experimental evaluation in Gong et al. (2021) employs the same performance metrics as before, but a slightly different evaluation protocol. Model training (and evaluation) is done in a 5-fold cross-validation setting, where the epoch with the best average accuracy across all five folds is chosen as the final model. The performance metrics given below are calculated by taking the predictions of all five folds at the chosen epoch and evaluating accuracy and calibration metrics jointly. While the accuracy calculated this way is equivalent to the average of all five folds, others are not, so this method results in a more realistic calculation of the calibration metrics.

The results are summarized in Tab. 6. On this dataset LoRA-Ensemble does not significantly outperform the Explicit Ensemble, but still matches its performance with much lower computational demands, see Appendix S. Accuracy is insignificantly lower, whereas calibration is slightly better in terms of ECE. We note that, remarkably, the weights used in the transformer modules and for creating patch embeddings were pre-trained on images rather than audio streams.

### A.3  CIFAR-100

Increasing the ensemble size of LoRA-Ensemble on CIFAR-100 improves classification accuracy but reduces calibration, as illustrated in Fig. 4. The reliability diagram in Fig. 5 highlights this behavior: networks with LoRA on CIFAR-100 are generally under-confident, with accuracy exceeding

Table 6: Model performance on the ESC-50 dataset for the compared methods. Ensembles have 8 members due to memory limitations. Best score for each metric in **bold**, second-best underlined.

| Method | Accuracy (↑) | F1 (↑) | ECE (↓) | NLL (↓) | Brier (↓) |
|---|---|---|---|---|---|
| Single Network | $89.6 \pm 0.7$ | $89.5 \pm 0.7$ | $0.039 \pm 0.004$ | $0.410 \pm 0.020$ | $0.164 \pm 0.009$ |
| Single Net w/ LoRA | $88.0 \pm 0.3$ | $87.8 \pm 0.3$ | $0.043 \pm 0.004$ | $0.461 \pm 0.019$ | $0.186 \pm 0.005$ |
| MC Dropout | $89.4 \pm 0.3$ | $89.3 \pm 0.4$ | $0.087 \pm 0.005$ | $0.553 \pm 0.012$ | $0.176 \pm 0.005$ |
| Explicit Ensemble | $\mathbf{91.3} \pm 0.2$ | $\mathbf{91.2} \pm 0.3$ | $\underline{0.027} \pm 0.004$ | $\mathbf{0.322} \pm 0.004$ | $\mathbf{0.133} \pm 0.001$ |
| LoRA-Ensemble | $\underline{91.1} \pm 0.2$ | $\underline{90.8} \pm 0.2$ | $\mathbf{0.021} \pm 0.003$ | $\underline{0.328} \pm 0.004$ | $\underline{0.138} \pm 0.001$ |

predicted confidence. As observed by Rahaman & Thiery (2020), ensembling under-confident models can exacerbate this discrepancy, leading to poorer calibration metrics.

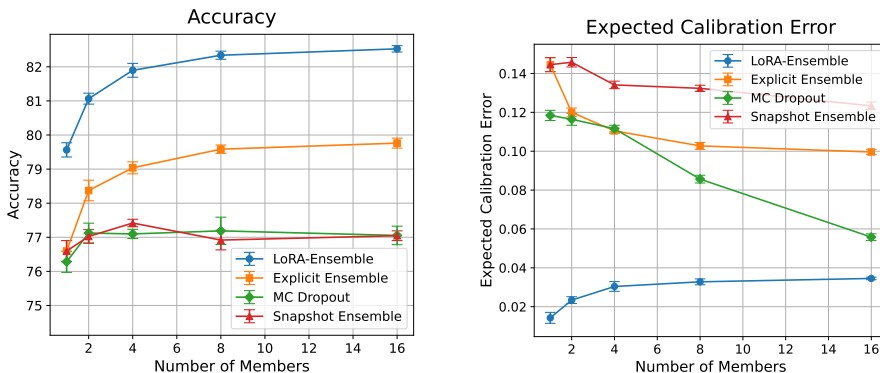

Figure 4: Accuracy and Expected Calibration Error on CIFAR-100, with different ensemble sizes.

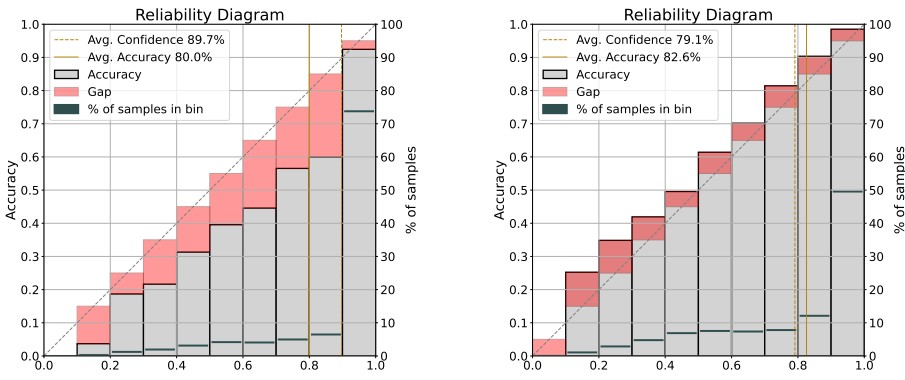

Figure 5: Reliability diagrams for Explicit Ensemble (left) and LoRA-Ensemble (right) with 16 members, on CIFAR-100.

### A.4 HAM10000 LESION CLASSIFICATION

Classification accuracy and ECE for HAM10000 dataset are both graphed against ensemble size in Fig. 6. Again, LoRA-Ensemble outperforms all baselines for larger ensembles. In Fig. 7 the reliability diagrams for LoRA-Ensemble and an Explicit Ensemble with 16 members each on the HAM10000 dataset are shown. Here, the models are overconfident, further supporting our reasoning regarding the surprising behaviour of calibration with growing ensemble size in the case of CIFAR-100.

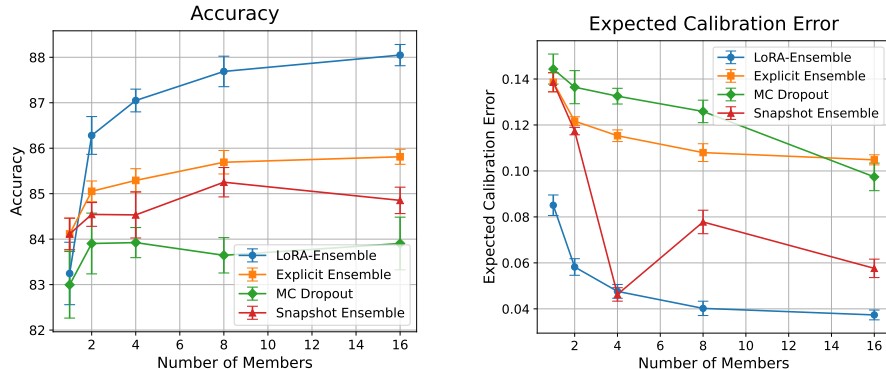

Figure 6: Accuracy and Expected Calibration Error on HAM10000, with different ensemble sizes.

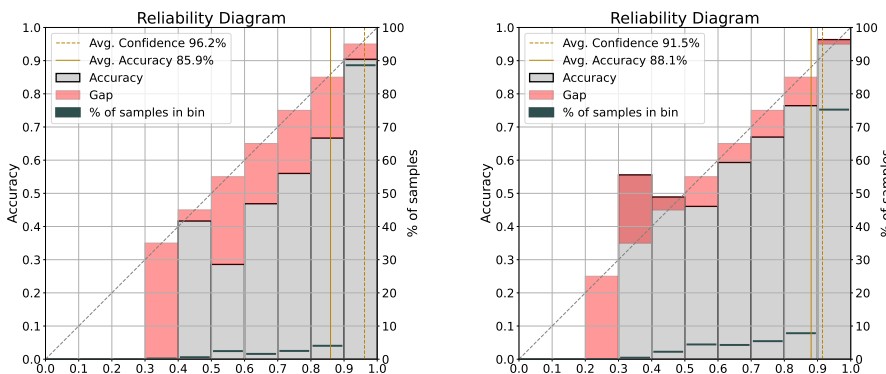

Figure 7: Reliability diagrams for Explicit Ensemble (left) and LoRA-Ensemble (right) with 16 members, on HAM10000.

### A.5 COMPARISON WITH A SINGLE, HIGH-RANK LORA NETWORK

We compare the proposed LoRA-Ensemble method with an additional baseline: a single high-rank LoRA model configured to have the same total number of trainable LoRA parameters as the LoRA-Ensemble. This evaluation is conducted on the CIFAR-100 classification task to examine the relative effectiveness of ensembling versus increasing parameter capacity within a single model.

Notably, as shown in Tab. 7, the high-rank LoRA model underperforms compared to the low-rank LoRA model. This result indicates that the performance gains of the LoRA-Ensemble are not solely due to an increased number of trainable parameters but are instead attributable to the ensembling approach.

### A.6 INATURALIST 2017 LARGE-SCALE FINE-GRAINED IMAGE CLASSIFICATION

In Fig. 8, reliability diagrams for the iNat2017 dataset are shown, once for LoRA-Ensemble and once for an Explicit Ensemble, both with 4 members. One can clearly see the over-confidence of

Table 7: Model performance on the CIFAR-100 dataset for the compared methods. Ensembles have 16 members. Best score for each metric in **bold**, second-best underlined.

| Method | Rank | Trainable params. | Accuracy (↑) | F1 (↑) | ECE (↓) | NLL (↓) | Brier (↓) |
|---|---|---|---|---|---|---|---|
| Single Net w/ LoRA | 8 | 666'724 | 79.6 ± 0.2 | 79.4 ± 0.2 | **0.014** ± 0.003 | 0.671 ± 0.005 | 0.286 ± 0.003 |
| Single Net w/ LoRA | 128 | 9'514'084 | 77.0 ± 0.1 | 77.0 ± 0.1 | 0.080 ± 0.001 | 0.867 ± 0.007 | 0.332 ± 0.002 |
| LoRA-Ensemble | 8 | 10'667'584 | **82.5** ± 0.1 | **82.5** ± 0.1 | 0.035 ± 0.001 | **0.587** ± 0.001 | **0.253** ± 0.000 |

the Explicit model, and the much improved uncertainty calibration of LoRA-Ensemble at almost the same accuracy (49.6% vs. 49.3%, c.f. Tab. 3).

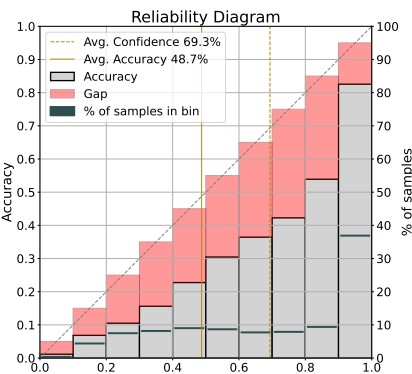 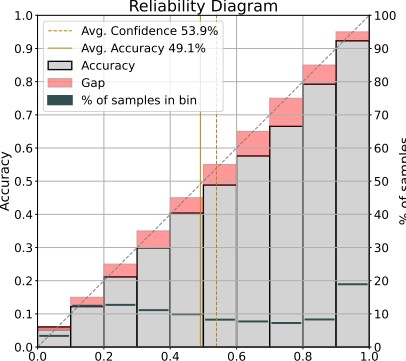

Figure 8: Reliability diagrams for Explicit Ensemble (left) and LoRA-Ensemble (right) with 4 members, on INat2017.

### A.7 SST-2 Language Modeling for Sentiment Classification

To further illustrate the generality of our method, we extend the evaluation to also include language processing (NLP). Indeed, we find that LoRA-Ensemble also handles this very different modality and estimates well-calibrated uncertainties for language data.

We chose the Stanford Sentiment Treebank 2 (SST-2) dataset Socher et al. (2013) for the experiment, a widely used binary sentiment classification benchmark, and part of the GLUE benchmark suite Wang et al. (2018). The model we use is the uncased BERT base model, which we fine-tune for SST-2.

Tab. 8 presents the performance on the SST-2 validation set. Among the methods, the LoRA-Ensemble shows strong overall performance: it achieves superior calibration in terms of negative log-likelihood (NLL), second-best ECE after Bayes-LoRA (with only a negligible difference), and outperforms all baselines including the Explicit Ensemble. The Explicit Ensemble holds only a marginal advantage of 0.5 percentage points in accuracy. In contrast, Monte Carlo Dropout improves calibration compared to single models but suffers from a substantial loss in accuracy, a pattern consistent with our other experiments and aligned with findings reported in the literature Li et al. (2023). A single LoRA-augmented model shows better calibration than a single model, but lags in accuracy. Bayes-LoRA achieves competitive uncertainty calibration, obtaining the best ECE, but its NLL, Brier score, and accuracy are worse than those of both the Explicit Ensemble and the LoRA-Ensemble, and its accuracy is even lower than a single model, reflecting a trade-off where improved calibration comes at the expense of predictive performance. Refer to Appendix W for more details and discussion about Bayes-LoRA.

Table 8: Performance on the SST-2 validation dataset, evaluated using five different random seeds per model. Ensembles have 8 members. Best score for each metric in **bold**, second-best underlined.

| Method | Accuracy (↑) | F1 (↑) | ECE (↓) | NLL (↓) | Brier (↓) |
|---|---|---|---|---|---|
| Single Network | $92.5 \pm 0.2$ | $92.5 \pm 0.2$ | $0.064 \pm 0.003$ | $0.345 \pm 0.012$ | $0.136 \pm 0.003$ |
| Single Net w/ LoRA | $91.6 \pm 0.5$ | $91.6 \pm 0.5$ | $0.059 \pm 0.005$ | $0.292 \pm 0.016$ | $0.148 \pm 0.008$ |
| MC Dropout | $84.9 \pm 1.2$ | $84.9 \pm 1.3$ | $0.061 \pm 0.004$ | $0.364 \pm 0.020$ | $0.223 \pm 0.015$ |
| Bayes-LoRA | $90.7 \pm 0.3$ | $90.7 \pm 0.3$ | $\mathbf{0.036} \pm 0.003$ | $0.247 \pm 0.005$ | $0.139 \pm 0.003$ |
| Explicit Ensemble | $\mathbf{93.2} \pm 0.2$ | $\mathbf{93.2} \pm 0.2$ | $0.047 \pm 0.002$ | $\underline{0.234} \pm 0.004$ | $\mathbf{0.112} \pm 0.003$ |
| LoRA-Ensemble | $\underline{92.7} \pm 0.2$ | $\underline{92.7} \pm 0.2$ | $\underline{0.038} \pm 0.003$ | $\mathbf{0.208} \pm 0.007$ | $\underline{0.114} \pm 0.002$ |

### A.8 ROBUSTNESS TO DISTRIBUTION SHIFTS: CIFAR-10-C AND CIFAR-100-C

Despite primarily evaluating the LoRA-Ensemble on in-distribution tasks, we also assess its robustness to out-of-distribution (OOD) inputs. A critical challenge arises when a model encounters data at test time that differs from the training distribution. If the model then produces poorly calibrated uncertainty estimates, this can lead to unsafe or unreliable predictions (Hendrycks & Dietterich, 2019).

To examine this, we evaluate our method on the CIFAR-10-C and CIFAR-100-C benchmark datasets. These datasets apply 19 distinct corruption types at five severity levels to the original CIFAR-10 and CIFAR-100 test sets (Hendrycks & Dietterich, 2019), introducing controlled distribution shifts, similar to prior work (Ovadia et al., 2019). For this evaluation, we use pretrained models trained on the clean datasets with minimal data augmentation (only rotations), and assess both predictive performance and calibration.

Fig. 9 and 10 present the results. It is evident that the LoRA-Ensemble outperforms the other methods, maintaining relatively low ECE scores even under high levels of distribution shift.

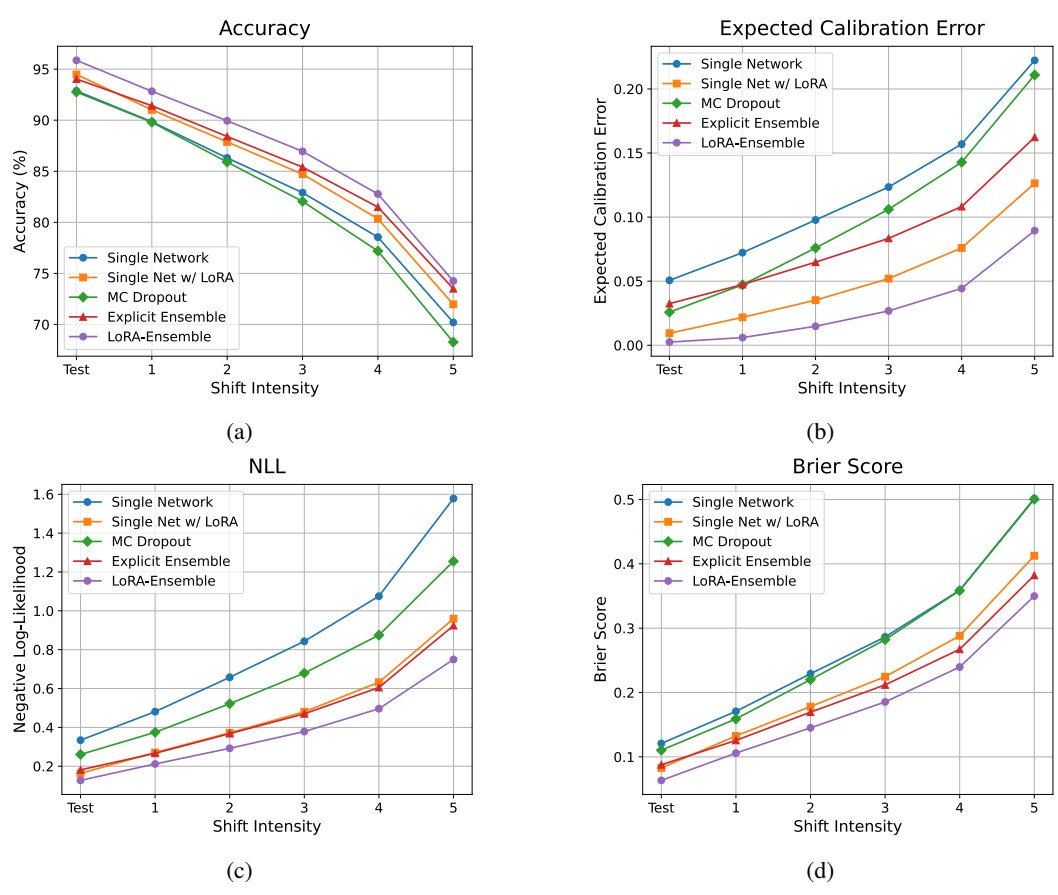

(a)                                    (b)

(c)                                    (d)

Figure 9: LoRA-Ensemble evaluated under varying levels of distribution shift on the CIFAR-10-C dataset. Each ensemble consists of 16 members. "Test" refers to the original CIFAR-10 test set, while the corrupted sets include test images subjected to 19 different augmentations at multiple severity levels, introducing distribution shifts.

### A.9 COMPUTATIONAL COST

In addition to evaluating classification performance and calibration, we assess the computational cost in terms of parameters, training time and inference time. The required resources are presented in Tab. 9.

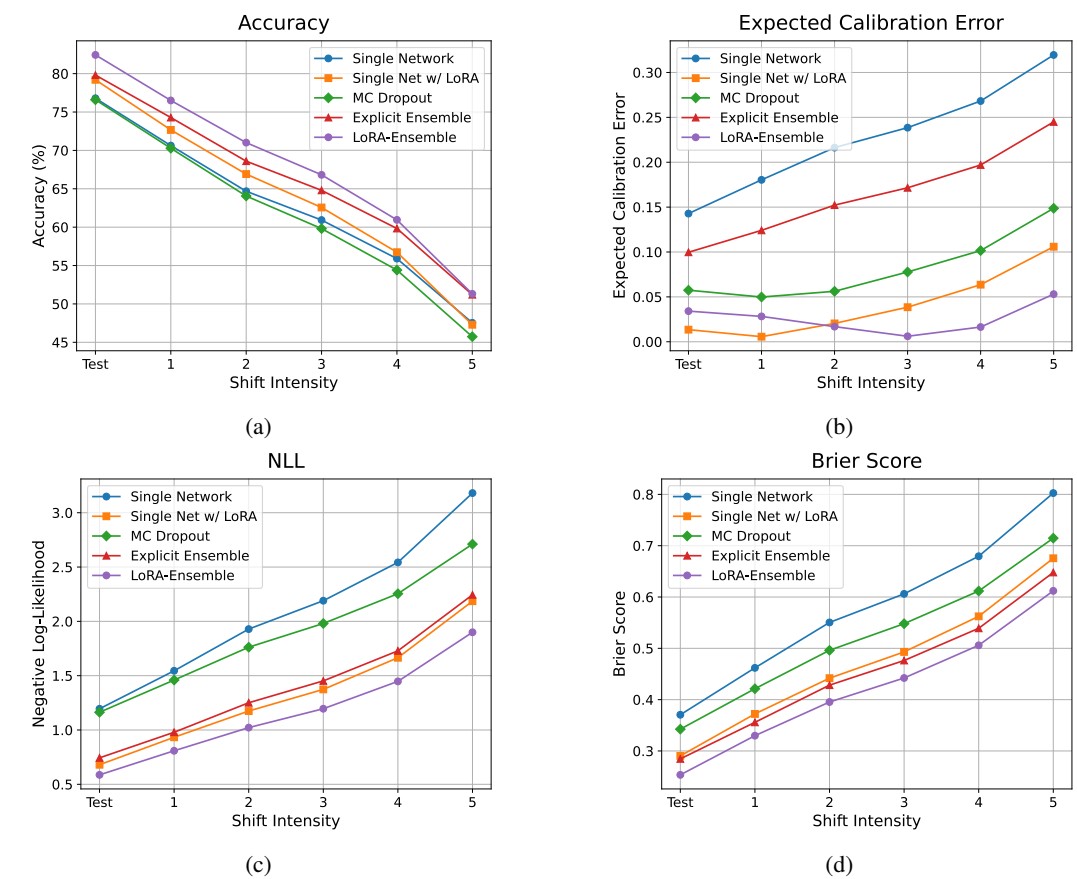

Figure 10: LoRA-Ensemble evaluated under varying levels of distribution shift on the CIFAR-100-C dataset. Each ensemble consists of 16 members. "Test" refers to the original CIFAR-100 test set, while the corrupted sets include test images subjected to 19 different augmentations at multiple severity levels, introducing distribution shifts.

Table 9: Parameter counts and computation times for an Explicit Ensemble of 16 ViT models and the corresponding LoRA-Ensemble. Training time is the average duration for one epoch on CIFAR-100, with batch size 32. Inference time is the average duration of a forward pass, with batch size 1.

| Method | Parameter overhead | Training time [s] | Inference time [ms] |
|---|---|---|---|
| Explicit Ensemble | $16 \times 87M$ | $16 \times 139$ | $16 \times 4.6$ |
| LoRA-Ensemble | $1.12 \times 87M$ | 1108 | 22.7 |

The total *number of parameters* is reported for an ensemble of 16 members, and matrices $A$ and $B$ with rank 8 when using LoRA. Choosing a different rank will slightly alter the parameter count. In many cases a lower rank may suffice, cf. Hu et al. (2021). All times were measured on a single NVIDIA Tesla A100-80GB GPU. *Training time* is given as the average wall clock time per training epoch on CIFAR-100, with 16 ensemble members. *Inference time* is computed as the average time for a single forward pass for a CIFAR-100 example, with batch size 1. The forward pass for the Explicit Ensemble processes the members sequentially[1], see also Appendix L. Hence, we calculate the average time needed for one member and multiply it by 16. It is evident that the proposed method uses significantly fewer parameters and less memory. LoRA-Ensemble also trains faster, and speeds up inference more than 3 times.

---

[1]Speed comparisons only make sense with the same resources. With sufficiently many GPUs any ensemble method can be parallelized by instantiating explicit copies of different members on separate GPUs.

We point out that, with our current implementation, the runtime comparisons are still indicative. It turns out that PyTorch's vectorized map (vmap) has a large one-time overhead that is only amortized when using large ensembles, while small ensembles are slowed down. Practical ensemble sizes will benefit when implemented in a framework that supports just-in-time compilation, like JAX.

# B  LoRA-Ensemble's Generalization to Varying Model Sizes

Building upon our existing experiments with the HAM10000 dataset, we extended our analysis to include different backbone architectures with varying numbers of parameters. Specifically, we utilized various DeiT models pre-trained with distillation, as described by Touvron et al. (2020). The results are presented in Table 10. Notably, the DeiT Base-32 model is the same as the ViT Base-32 model.

In the small parameter regime (Tiny-16, Small-16), the addition of a single LoRA module did not consistently enhance calibration compared to using a single model. This observation contrasts with our findings in most other experiments. However, in the larger parameter regime (ViT Base-32), incorporating even a single LoRA module significantly improved calibration. Increasing the number of ensembles in the LoRA-Ensemble not only boosted accuracy but also enhanced calibration, enabling it to match the performance of an Explicit Ensemble in both parameter regimes.

Last but not least, as the number of parameters in the backbone architecture increased, the superiority of the LoRA-Ensemble over the Explicit Ensemble in terms of both accuracy and calibration became more pronounced. This trend indicates that as backbone size grows, the advantages of LoRA-Ensemble become increasingly dominant.

Overall, the results demonstrate that the LoRA-Ensemble not only transfers successfully to a different backbone architecture (DeiT versus ViT) but also remains effective across varying parameter regimes.

Table 10: Performance metrics on the HAM10000 dataset for different Vision Transformer architectures. Ensembles have 16 members. The top two results for each metric are highlighted: **bold** for the best, underlined for the second best.

| Arch. | Method | # Params. | Accuracy (↑) | F1 (↑) | ECE (↓) | NLL (↓) | Brier (↓) |
|---|---|---|---|---|---|---|---|
| DeiT Tiny-16 | Single Net | 5 M | 89.0 ± 0.3 | 79.0 ± 0.4 | 0.096 ± 0.003 | 0.909 ± 0.037 | 0.202 ± 0.005 |
| | Single Net w/ LoRA | | 84.5 ± 0.8 | 71.6 ± 1.5 | 0.074 ± 0.003 | 0.542 ± 0.017 | 0.237 ± 0.009 |
| | Explicit Ensemble | | **90.4** ± 0.3 | **81.4** ± 0.4 | 0.069 ± 0.004 | 0.340 ± 0.006 | **0.142** ± 0.002 |
| | LoRA-Ensemble | | 88.9 ± 0.4 | 80.6 ± 0.2 | **0.025** ± 0.003 | **0.325** ± 0.004 | 0.164 ± 0.002 |
| DeiT Small-16 | Single Net | 22 M | 89.6 ± 0.4 | 79.0 ± 0.5 | 0.093 ± 0.003 | 0.876 ± 0.032 | 0.191 ± 0.007 |
| | Single Net w/ LoRA | | 86.3 ± 0.5 | 76.8 ± 1.0 | 0.100 ± 0.007 | 0.731 ± 0.053 | 0.234 ± 0.010 |
| | Explicit Ensemble | | **91.5** ± 0.1 | 82.4 ± 0.2 | 0.061 ± 0.002 | 0.318 ± 0.003 | **0.130** ± 0.001 |
| | LoRA-Ensemble | | 90.4 ± 0.1 | **82.8** ± 0.4 | **0.047** ± 0.002 | **0.292** ± 0.002 | 0.144 ± 0.001 |
| DeiT Base-32 | Single Net | 86 M | 84.1 ± 0.3 | 71.4 ± 0.7 | 0.139 ± 0.004 | 1.138 ± 0.040 | 0.291 ± 0.009 |
| | Single Net w/ LoRA | | 83.2 ± 0.7 | 70.7 ± 1.3 | 0.085 ± 0.004 | 0.569 ± 0.027 | 0.256 ± 0.011 |
| | Explicit Ensemble | | 85.8 ± 0.2 | 74.6 ± 0.4 | 0.105 ± 0.002 | 0.536 ± 0.007 | 0.218 ± 0.002 |
| | LoRA-Ensemble | | **88.0** ± 0.2 | **78.3** ± 0.6 | **0.037** ± 0.002 | **0.342** ± 0.003 | **0.175** ± 0.002 |

# C  Hyperparameter Selection and Sensitivity Analysis: LoRA Rank

The main hyper-parameter introduced by adding LoRA is the rank of the low-rank decomposition (i.e., the common dimension of the matrices $A$ and $B$). Varying that rank modulates the complexity of the model for the learning task. We have empirically studied the relationship between rank, accuracy, and Expected Calibration Error. Here we show results for HAM10000 and CIFAR-100 dataset.

On HAM10000 we observe a clear trade-off between accuracy and calibration, Fig. 11a. With increasing rank the classification accuracy increases while the calibration deteriorates, in other words, one can to some degree balance predictive accuracy against uncertainty calibration by choosing the

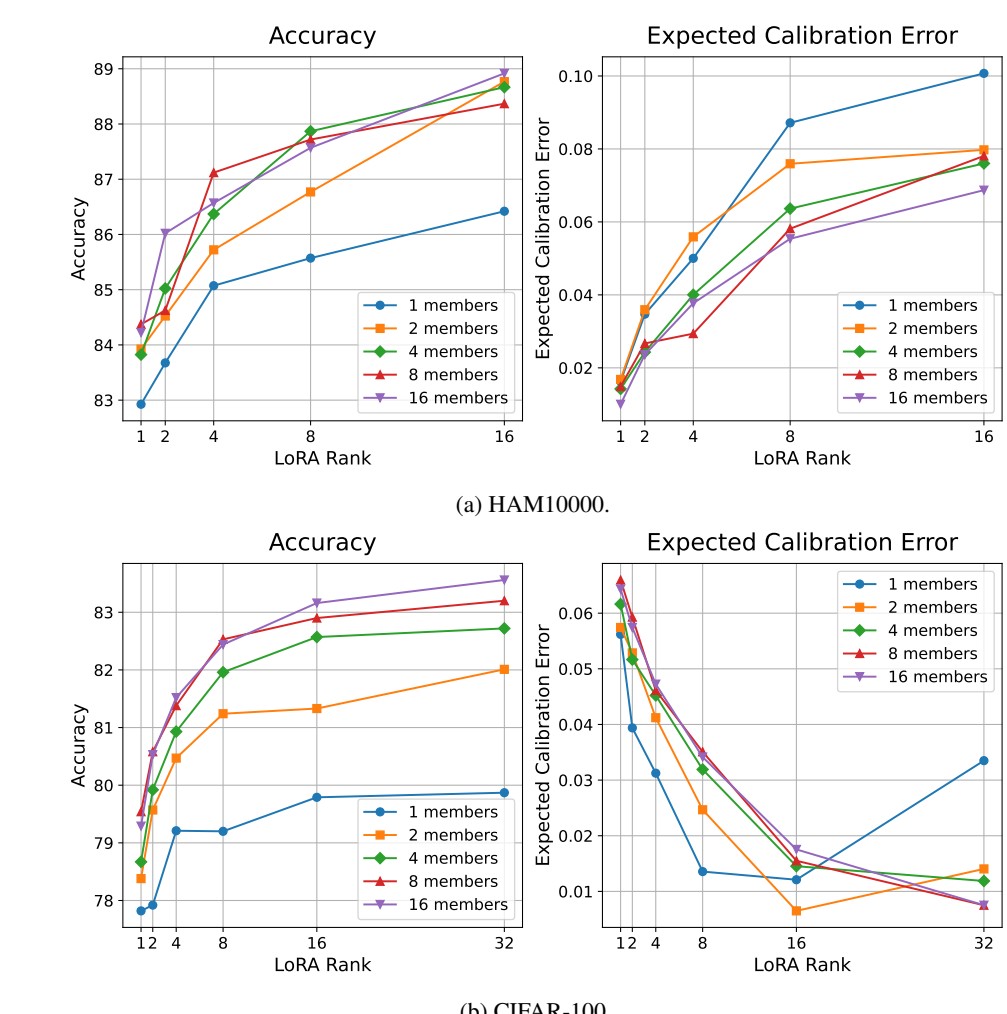

(a) HAM10000.

(b) CIFAR-100

Figure 11: Impact of LoRA rank on accuracy and ECE.

rank. Our focus in this work is on model calibration. We therefore generally choose the rank to favor calibration, even at the cost of slightly lower classification accuracy.

For the CIFAR-100 dataset, our evaluation of LoRA-Ensemble shows both increased accuracy and improved calibration with increasing rank within the studied range. These findings are illustrated in Fig. 11b.

This observation aligns with the findings of Rahaman & Thiery (2020), as LoRA-Ensemble continues to exhibit under-confidence even at higher ranks. Increasing model complexity enhances confidence, thereby improving calibration. However, at rank 32, the calibration of a single network augmented with LoRA begins to deteriorate, suggesting that a critical boundary has been reached. Beyond this point, the parameter space becomes insufficiently constrained, leading to effects similar to those observed by Guo et al. (2017).

At higher ranks, accuracy plateaus while memory demand increases linearly with $\mathcal{O}(d)$ and $\mathcal{O}(k)$ for $A \in \mathbb{R}^{r \times d}$ and $B \in \mathbb{R}^{k \times r}$ respectively, where $d$ and $k$ are the dimensions of the pre-trained weight matrix $W_0 \in \mathbb{R}^{k \times d}$. Consequently, we selected rank 8 for our CIFAR-100 experiments.

Overall, the rank serves as the primary control of LoRA's expressive capacity. While larger values tend to improve performance on more complex datasets (e.g., rank 64 for INaturalist), excessively large choices (e.g., $\geq 256$) suppress the distinctive dimension-learning behavior of LoRA-Ensemble, resulting not only in diminishing returns but in some cases an actual decline in accuracy. In practice,

we find that a small sweep over $\{4, 8, 16, 32, 64\}$ on a held-out set is typically sufficient to identify a near-optimal rank.

# D WEIGHT SPACE ANALYSIS: LoRA-ENSEMBLE VERSUS EXPLICIT ENSEMBLE

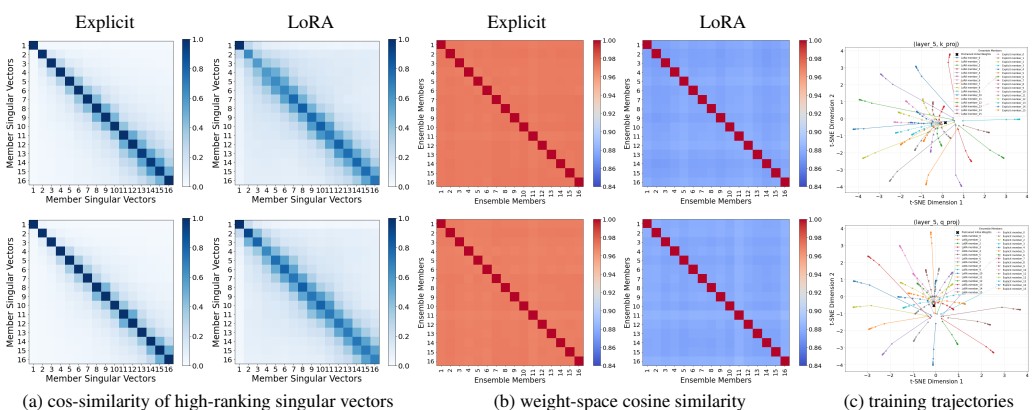

(a) cos-similarity of high-ranking singular vectors    (b) weight-space cosine similarity    (c) training trajectories

Figure 12: Weight space analysis of LoRA-Ensemble vs. Explicit Ensemble: The first row represents key matrices, while the second row represents query matrices.

This section expands on Sec. 4, which examines the diversity of ensemble members in function and weight space for LoRA-Ensemble and Explicit Ensemble, showing that LoRA-Ensemble exhibits greater diversity in both spaces. While Sec. 4 focuses on value projection matrices due to their role in learned representations, this section examines query and key projection matrices, too. In Fig. 12, we observe that LoRA-Ensemble achieves greater diversity in query and key projection matrices, similar to the diversity observed in value projection matrices (Fig. 3).

Using Singular Value Decomposition (SVD), a weight matrix $W \in \mathbb{R}^{m \times n}$ is decomposed as:

$$W = U\Sigma V^\top,$$

where $U \in \mathbb{R}^{m \times m}$ and $V \in \mathbb{R}^{n \times n}$ are orthonormal matrices representing rotational components, and $\Sigma \in \mathbb{R}^{m \times n}$ is a diagonal matrix of singular values capturing the scaling effect. Singular vectors linked to larger singular values highlight key transformations encoded by $W$.

In Fig. 13, we analyze the differences in weight updates between ensemble methods by computing the Singular Value Decomposition (SVD) of pre-trained and trained weights for ensemble members. Singular vectors corresponding to the top singular values (16 are shown) are extracted and compared using cosine similarity to evaluate changes in the weight structure. These similarities are averaged across layers and ensemble members. The results highlight distinct parameter update patterns between LoRA-Ensemble and Explicit Ensemble. LoRA-Ensemble introduces new high-ranking singular vectors, referred to as "intruder dimensions" Shuttleworth et al. (2024), which are nearly orthogonal to the singular vectors of the pre-trained weights. The number of intruder dimensions depends on the LoRA rank. This effect is particularly pronounced in the value projection matrices, which aligns with their strong association with learned representations. In contrast, Explicit Ensemble members tend to preserve a structure closely aligned with the spectral properties of the pre-trained weights. This alignment is especially evident in the key and query projection matrices, which exhibit a strong resemblance to the original spectral structure.

We further analyze the $B \cdot A$ matrices learned by different ensemble members. Due to their random initialization, these matrices explore diverse directions in weight space. In Fig. 14, we plot the largest eigenvalues of these matrices (with only four non-zero eigenvalues as the LoRA rank is set to 4) and the similarity between the corresponding eigenvectors across ensemble members. The similarities are averaged over layers and member pairs. The results show that while the eigenvalues across members follow a similar trend, the eigenvectors are largely uncorrelated. This indicates

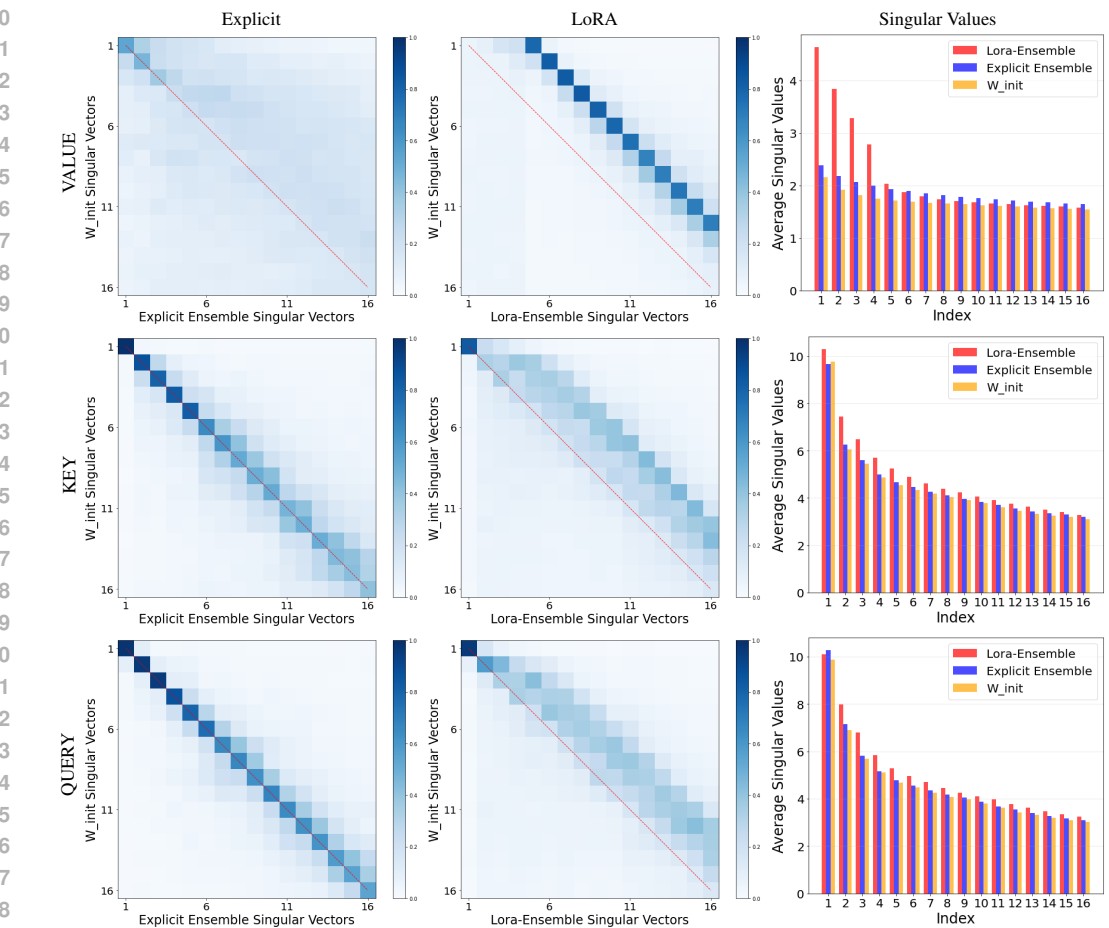

Figure 13: Cosine similarity of top singular vectors (and associated singular values) between initial pre-trained and final trained weights, averaged over layers and ensemble members.

that ensemble members explore different regions of weight space while maintaining similar overall transformations. The shared eigenvalue trends suggest consistent semantic contributions across members, while the dissimilar eigenvectors highlight the diversity in their learned representations.

We plot the t-SNE visualizations for different layers in Fig. 15, capturing the evolution of weights during training. The visualizations include the initial pretrained weights, and for each ensemble member, we plot weights from epoch 5 to epoch 65 at 5-epoch intervals. The plots reveal that LoRA-Ensemble members exhibit broader convergence across the loss landscape in various layers, signifying diverse learning dynamics. Conversely, Explicit Ensemble members tend to remain closer to their initial weights, indicating reduced diversity throughout the training process.

# E CORRELATION ANALYSIS BETWEEN ENSEMBLE DIVERSITY AND PREDICTIVE PERFORMANCE

Prior work has shown that diversity across modes in weight space correlates with improved uncertainty estimates (Fort et al., 2019b; Izmailov et al., 2021). To investigate this phenomenon in the context of LoRA, we trained eight-member ensembles on the HAM10000 dataset using varying LoRA initialization gains.

We treat the LoRA initialization gain as a simple *diversity knob*: larger gains induce greater spread among the low-rank adapters. To quantify ensemble diversity, we compute the average pairwise correlation between the LoRA $V$ projection matrices of different ensemble members, averaged across

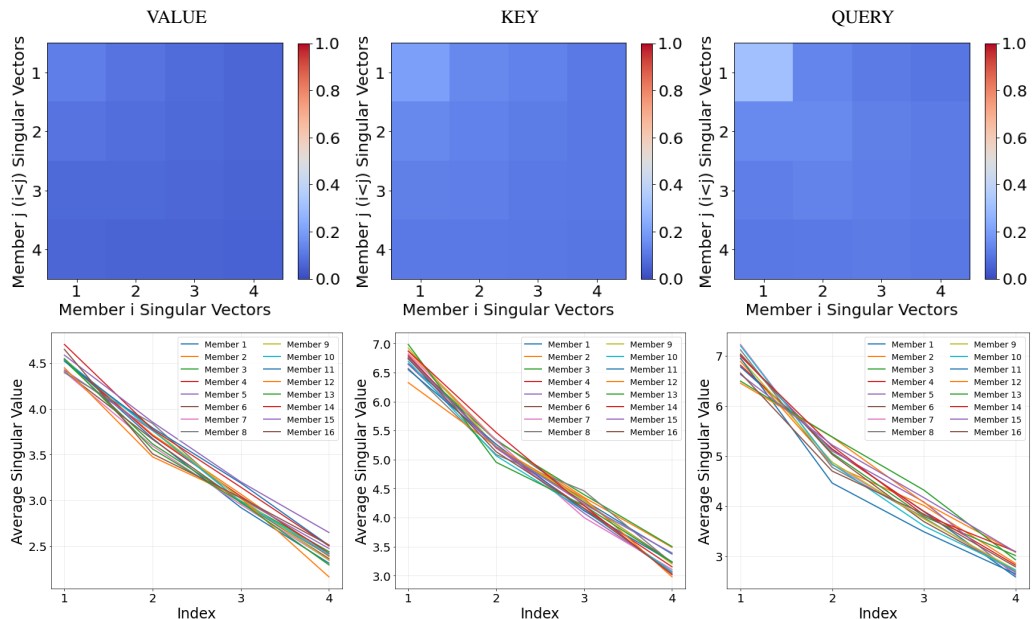

Figure 14: Cosine similarity of top singular vectors from $B \cdot A$ low-rank matrices (rank set to 4) between LoRA-Ensemble members, averaged over layers and all member pairs (first row), along with corresponding average singular values for different members (second row).

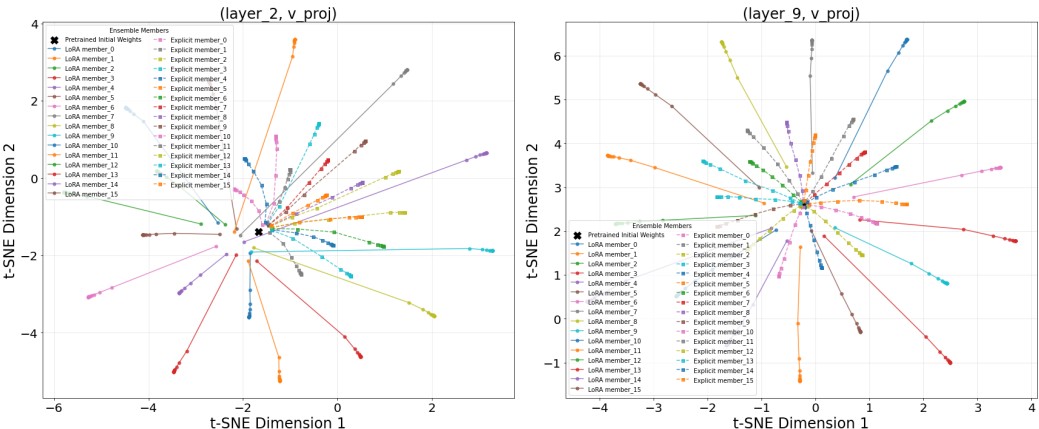

Figure 15: Training trajectories of ensemble members of LoRA-Ensemble and Explicit Ensemble.

all layers. The *diversity score* is then defined as

$$\text{Diversity score} = 1 - \text{average correlation}.$$

This metric captures how dissimilar the ensemble members are in terms of their learned parameters, with higher scores indicating less correlated (i.e., more diverse) adapters.

Tab. 11 summarizes the results. Both performance and calibration improve as diversity increases, up to a point. Beyond a certain threshold, additional diversity provides diminishing returns and eventually plateaus. Moreover, as shown in Fig. 14, excessively large gains degrade performance. From a Bayesian perspective, moderate gains behave like a well-chosen prior variance, encouraging the ensemble to explore distinct posterior modes. Excessively large gains, however, make the initial weights too diffuse, which can cause activation saturation, unstable gradients, suboptimal convergence, and ultimately higher variance and miscalibration. A sweep of gain values reveals a U-shaped

calibration curve: uncertainty estimates improve with increasing diversity until diversity becomes excessive, at which point performance and calibration degrade.

| Gain | Diversity Score | Accuracy | NLL |
|------|-----------------|----------|-----|
| 1 | 0.006 | 0.854 | 0.405 |
| 2 | 0.017 | 0.856 | 0.393 |
| 4 | 0.048 | 0.860 | 0.374 |
| 8 | 0.117 | 0.877 | 0.354 |
| 12 | 0.184 | 0.881 | 0.345 |
| 20 | 0.304 | 0.882 | 0.345 |
| 40 | 0.511 | 0.882 | 0.344 |

Table 11: Effect of LoRA initialization gain on ensemble diversity, accuracy, and calibration (measured by NLL) on HAM10000.

## F   JOINT TRAINING OF BACKBONE AND LORA-ENSEMBLE LAYERS ON INATURALIST

LoRA-Ensemble keeps the backbone weights frozen and trains only the low-rank matrices. To assess the impact of relaxing this constraint, we evaluated the method in a setting where the full backbone is also trainable. Results on the INat2017 dataset, see Tab. 12, show a substantial drop in both accuracy and calibration when the entire network is trained, although performance still surpasses that of a single model.

We hypothesize that enabling backbone training washes out the low-rank adjustments introduced by LoRA. With a frozen backbone, each model's unique low-rank matrices create intruder dimensions that yield diverse feature spaces. See Appendix D for more details. Once the backbone is trainable, those adjustments merge into the dominant spectral modes, causing all ensemble members to collapse into the same parameter region and behave like a single network. Similar behavior was observed for the Batch-Ensemble method, as discussed in Appendix I.

Table 12: Performance on the INat2017 dataset. 'full' indicates that the entire network, including the pre-trained backbone, is trainable. Ensembles consist of 4 members. Best score for each metric in **bold**, second-best underlined.

| Method | Accuracy (↑) | F1 (↑) | ECE (↓) | NLL (↓) | Brier (↓) |
|--------|--------------|--------|---------|---------|-----------|
| Single Network | 42.6 | 37.8 | 0.293 | 1.054 | 0.207 |
| Single Net w/ LoRA | 47.7 | 43.1 | 0.096 | 0.662 | 0.166 |
| Single Net w/ LoRA (full) | 42.8 | 38.0 | 0.271 | 0.958 | 0.201 |
| LoRA-Ensemble | **49.3** | **44.1** | **0.045** | **0.610** | **0.160** |
| LoRA-Ensemble (full) | 44.0 | 39.4 | 0.249 | 0.886 | 0.193 |

## G   PLACEMENT OF LORA-ENSEMBLE MODULES AND SELECTION OF TRAINABLE WEIGHTS

Typically, LoRA is applied only to the weights in the multi-head attention module (i.e., the query, key, value, and output projections), as demonstrated in the original LoRA paper Hu et al. (2021). We acknowledge that, due to the modular nature of transformer architectures, LoRA layers can also be inserted into the feedforward MLP blocks. While this alternative placement has been shown in prior work to improve performance on certain datasets, it may also lead to reduced robustness and lower overall performance Fomenko et al. (2024). Moreover, the projection matrices in the MLP blocks typically have significantly higher dimensionality, often four times larger than those in the attention layers. As a result, this placement introduces a substantial number of additional parameters, which

can increase memory consumption. This effect is especially pronounced when ensemble members are executed in parallel using vectorized mapping rather than sequential execution.

Table 13 presents the results on the HAM10000 dataset. Notably, adding LoRA-Ensemble layers to the MLP blocks leads to improved accuracy, but at the cost of poorer calibration performance. Additionally, when the embedding layers of the Vision Transformer (ViT) are also trained alongside the low-rank matrices for the purpose of patch feature extraction, we observe a marked drop in classification accuracy. This performance degradation can be attributed to the substantial number of additional parameters introduced in the early stages of the model, which are by design an order of magnitude larger than in the subsequent LoRA layers, potentially leading to suboptimal training dynamics.

Finally, we observe that assigning a separate classification head to each LoRA-Ensemble member yields further performance gains. However, we also note that this design choice can be omitted in favor of improved parameter efficiency, depending on the application constraints.

Table 13: Ablation Study. Investigates the placement of LoRA-Ensemble layers and additional trainable components on the HAM10000 dataset. Ensembles consist of 8 members. Best score for each metric in **bold**, second-best underlined.

| LoRA-Ensemble Config. | Extra Trainable Layers | Trainable Params. | Accuracy (↑) | F1 (↑) | ECE (↓) | NLL (↓) | Brier (↓) |
|---|---|---|---|---|---|---|---|
| Multi-head attention | Cls. head | 2'364'679 | 87.5 | 77.7 | 0.041 | 0.365 | 0.187 |
| Multi-head attention | Cls. head + tokenizer | 4'724'743 | 84.6 | 73.8 | **0.025** | 0.422 | 0.217 |
| Multi-head attention + MLP | Cls. head | 5'313'799 | **90.1** | **80.9** | 0.077 | 0.383 | **0.157** |
| Multi-head attention + MLP | Cls. head + tokenizer | 7'673'863 | 87.4 | 77.1 | 0.083 | 0.438 | 0.192 |
| Multi-head attention | Full backbone | 90'114'055 | 85.2 | 73.3 | 0.126 | 1.000 | 0.264 |
| Multi-head attention | Ensemble cls. head | 2'402'360 | 88.0 | 78.0 | 0.036 | **0.347** | 0.179 |

# H    LORA-ENSEMBLE FOR CNNS

We extend LoRA Ensemble to convolutional neural networks (CNN) by applying it to a ResNet-18 backbone with an ensemble of four members. We mainly follow the original Batch-Ensemble Wen et al. (2020) implementation. For detailed experimental settings, see Turkoglu et al. (2022). Table 14 reports the CIFAR-100 results. LoRA-Ensemble achieves the second-best performance among implicit ensembling methods, behind only FiLM-Ensemble, but it does not match its efficacy on transformer architectures. As discussed in the main text and in the Appendix V, this gap stems from the fundamentally different computational structures of transformers compared with MLPs and CNNs, which makes direct adaptation of techniques between these domains challenging.

Table 14: Performance on the CIFAR-100 dataset for CNN architecture. Ensembles have 4 members and Resnet-18 is used as a backbone. For implicit ensemble methods, the best score for each metric in **bold**, second-best underlined.

| Method | Accuracy (↑) | ECE (↓) |
|---|---|---|
| Single Network | $78.0 \pm 0.4$ | $0.046 \pm 0.001$ |
| Deep Ensemble | $81.6 \pm 0.3$ | $0.041 \pm 0.002$ |
| MC-Dropout | $75.5 \pm 0.6$ | $0.064 \pm 0.003$ |
| MIMO | $48.0 \pm 2.6$ | $0.083 \pm 0.017$ |
| Masksemble | $72.5 \pm 0.5$ | $0.075 \pm 0.004$ |
| FiLM-Ensemble | $\mathbf{79.4} \pm 0.2$ | $\mathbf{0.038} \pm 0.000$ |
| Batch-Ensemble | $77.7 \pm 0.1$ | $0.052 \pm 0.002$ |
| LoRA-Ensemble | $\underline{78.4} \pm 0.2$ | $\underline{0.048} \pm 0.001$ |

# I BATCH-ENSEMBLE

## I.1 IMPLEMENTATION

Probably the closest method to LoRA-Ensemble is Batch-Ensemble introduced in Wen et al. (2020). Batch-Ensemble was originally developed for MLPs, but we extend it to self-attention networks as an implicit ensemble baseline. The methodology draws inspiration from our development of LoRA-Ensemble, as the two implementations share many similarities. The primary difference lies in the parametrization of ensemble members. For each projection matrix (query, key, value, and output), we define one shared full-rank trainable matrix initialized with the pre-trained weights of the base network, along with two additional trainable vectors, $r$ and $s$, which are specific to each ensemble member. The projection matrix for ensemble member $i$ is defined as:

$$W_i = W_{\text{shared}} \circ r_i s_i^T, \tag{5}$$

where $W_{\text{shared}}$ is the shared trainable matrix, and $\circ$ denotes element-wise multiplication. Within each transformer block, a unique forward pass is computed for each ensemble member $i$:

$$h_i = W_i x, \tag{6}$$

resulting in $N$ different predictions $T_{\theta_i}(X)$ for a given input $X$. The final ensemble prediction is obtained by averaging the individual predictions:

$$\mathbb{E}[Y|X] = \frac{1}{N} \sum_{i=1}^{N} T_{\theta_i}(X). \tag{7}$$

The forward pass for the Batch-Ensemble layer with shared weights is implemented as shown in Listing 1.

Listing 1: Pytorch forward pass for Batch-Ensemble layer

```python
def forward(self, x):
    """
    Forward pass for the Batch-Ensemble layer
    """
    # Step 1: Compute the ensemble member-specific weights
    r = self.r.weight  # Shape: [1, dim]
    s = self.s.weight  # Shape: [out_dim, 1]
    W_rs = s @ r       # Shape: [out_dim, dim]

    # Step 2: Combine with the shared weight
    W_combined = self.shared_w * W_rs  # Element-wise multiplication

    # Step 3: Compute the output for a specific ensemble member
    out = x @ W_combined.T  # x must have shape [batch_size, dim]

    return out
```

The $r$ and $s$ vectors are initialized from a Gaussian distribution centered around 1, specifically $r, s \sim \mathcal{N}(1, \sigma^2)$, where $\sigma^2$ controls the variance. We empirically set $\sigma^2 = 0.02$. This initialization ensures that at the beginning of the training, the combined projection matrix for each ensemble member remains close to the pre-trained weights of the shared matrix, preventing disruption of learned pre-trained weights. The implementation and training details followed the LoRA-Ensmeble approach; for details, refer to L and M.

## I.2 WHY LORA-ENSEMBLE OUTPERFORMS BATCH-ENSEMBLE

Both LoRA-Ensemble and Batch-Ensemble leverage shared weights with member-specific low-rank modifications to enable efficient ensembling. The key difference lies in their parameterization: LoRA-Ensemble uses additive low-rank updates, while Batch-Ensemble applies element-wise multiplicative scaling. Despite the conceptual similarity between the two methods, Batch-Ensemble performs significantly worse than LoRA-Ensemble in both accuracy and calibration, as demonstrated in Tab. 1 and Tab. .2 This performance gap persists even when applied to non–self-attention

architectures such as convolutional neural networks, which were the original target application of Batch-Ensemble, as shown in Tab. 14.

To clarify this difference, we examine the gradients of the member-specific parameters. For LoRA-Ensemble, the layer output is:

$$h_i = W_{\text{shared}} \cdot x + B_i A_i x,$$

with gradients:

$$\frac{\partial \mathcal{L}}{\partial B_i} = \delta \cdot A_i \cdot x, \tag{8}$$

$$\frac{\partial \mathcal{L}}{\partial A_i} = \delta \cdot B_i \cdot x, \tag{9}$$

where $\delta = \frac{\partial \mathcal{L}}{\partial h_i}$.

For Batch-Ensemble, the output is:

$$h_i = (W_{\text{shared}} \odot r_i s_i^T) x,$$

with gradients:

$$\frac{\partial \mathcal{L}}{\partial s_i} = \delta \cdot (W_{\text{shared}} \odot r_i) \cdot x, \tag{10}$$

$$\frac{\partial \mathcal{L}}{\partial r_i} = \delta \cdot (W_{\text{shared}} \odot s_i^T) \cdot x. \tag{11}$$

In Batch-Ensemble, the gradient updates for $r_i$ and $s_i$ are directly scaled by the shared weights $W_{\text{shared}}$, which can constrain the learning dynamics and reduce the independence of ensemble members. This scaling introduces sensitivity to the magnitude and sparsity of $W_{\text{shared}}$, potentially limiting the diversity of the ensemble.

We define **Batch-Ensemble++** by modifying the original Batch-Ensemble algorithm, replacing the point-wise multiplication operation with an addition operation as follows:

$$W_i = W_{\text{shared}} + r_i s_i^T. \tag{12}$$

In this case, the r and s vectors are also initialized from a Gaussian distribution but centered around 0.

We compare the performance of Batch-Ensemble++, the original Batch-Ensemble, and LoRA-Ensemble in Tab. 15. Batch-Ensemble++ significantly outperforms the original Batch-Ensemble in both accuracy and uncertainty calibration. However, its performance does not reach that of LoRA-Ensemble.

We attribute this performance gap to the following key differences between the methods:

- **Limited Expressiveness:** Batch-Ensemble restricts its ensemble-specific parameters to rank-1 matrices, inherently limiting the expressive power of individual ensemble members.
- **Coupled Learning Dynamics:** In Batch-Ensemble, the shared pre-trained matrix $W_{\text{shared}}$ is not kept frozen. This design choice can disrupt the learned pre-trained weights and may restrict the ability of the ensemble-specific parameters $r$ and $s$ to learn a sufficiently diverse set of features. A similar effect was observed in LoRA-Ensemble when the backbone was also updated during training; see the Appendix F for details.
- **Initialization Variations:** Differences in parameter initialization may also contribute to the performance gap.

## J   LORA-ENSEMBLE FINE-TUNED ON THE SAME DATASET AS THE BACKBONE MODEL

While our study explicitly focused on transfer learning setups, we also explored how LoRA-Ensemble can be applied when the backbone is trained on the same dataset. To this end, we initialized the LoRA ensemble with weights from a single network trained for 65 epochs on HAM10000,

Table 15: Model performance on the CIFAR-10 dataset for the compared methods. Ensembles have 4 members. Best score for each metric in **bold**, second-best underlined.

| Method | Accuracy (↑) | F1 (↑) | ECE (↓) | NLL (↓) | Brier (↓) |
|---|---|---|---|---|---|
| Single Network | 92.8 | 92.8 | 0.051 | 0.333 | 0.120 |
| Batch-Ensemble | 88.5 | 88.5 | 0.046 | 0.345 | 0.171 |
| Batch-Ensemble++ | 91.7 | 91.7 | 0.033 | 0.260 | 0.128 |
| LoRA-Ensemble | **95.6** | **95.6** | **0.003** | **0.133** | **0.067** |

and subsequently fine-tuned it for one epoch without learning rate warmup. Fig. 16 presents the results of fine-tuning a LoRA-Ensemble with rank 2 for a single epoch. It is evident that even in this scenario, the LoRA-Ensemble improves both performance and calibration with minimal computational overhead. We also highlight that alternative methods, such as explicit ensembling, are not directly applicable in this context.

Table 16: LoRA-Ensemble performance when it is fine-tuned on a pre-trained dataset. The HAM10000 dataset is used, and the ensemble consists of 8 members. The backbone is identical to that of the Single Network, which is fine-tuned for one epoch. Best score for each metric in **bold**.

| Method | Accuracy (↑) | F1 (↑) | ECE (↓) | NLL (↓) | Brier (↓) |
|---|---|---|---|---|---|
| Single Network | $84.1 \pm 0.3$ | $71.4 \pm 0.7$ | $0.139 \pm 0.004$ | $1.138 \pm 0.040$ | $0.291 \pm 0.009$ |
| Single Net w/ LoRA | $83.2 \pm 0.7$ | $70.7 \pm 1.3$ | $0.085 \pm 0.004$ | $0.569 \pm 0.027$ | $0.256 \pm 0.011$ |
| LoRA-Ensemble (finetuned for 1 epoch) | **84.8** | **72.2** | **0.059** | **0.514** | **0.238** |

## K  POST-HOC TEMPERATURE SCALING FOR MODEL CALIBRATION

Temperature scaling is a simple yet effective post-hoc calibration method used to improve the confidence of probabilistic models Guo et al. (2017). It rescales the logits of a trained model by a scalar parameter $T > 0$ (the temperature). Given logits $\mathbf{z}$, the calibrated probabilities $\hat{p}_i$ for class $i$ are computed as:

$$\hat{p}_i = \frac{\exp(z_i/T)}{\sum_j \exp(z_j/T)}. \tag{13}$$

Here, $T = 1$ corresponds to no scaling, and $T > 1$ reduces overconfidence by softening the logits.

To assess the impact of temperature scaling on calibration, we conducted experiments on CIFAR-100 with varying temperature values, as shown in Tab. 17. For each method, the model parameters were fixed, and the effect of different temperatures on calibration was evaluated. We observe that calibration can be improved across all methods, with the exception of the single network with LoRA, which does not require temperature scaling.

As discussed in Section 3, LoRA-Ensemble is under-confident on CIFAR-100, as evidenced by the optimal temperature being less than 1 for this method.

## L  IMPLEMENTATION OF LORA-ENSMEBLE

In practice, our LoRA-Ensemble is implemented by replacing the respective linear layers ($W_q$, $W_k$, $W_v$, and $W_o$) in the pre-trained model architecture with custom LoRA modules.

As a backbone for experiments with image datasets, we employ a Vision Transformer (ViT) model Dosovitskiy et al. (2020). The chosen architecture is the *base* variant with patch size $32 \times 32$ as defined in Dosovitskiy et al. (2020). We load the weights from `torchvision`, which were trained on ImageNet-1k Deng et al. (2009), using a variant of the training recipe from Touvron et al. (2020), for details refer to their documentation.

Table 17: Model performance on the CIFAR-100 dataset with different temperature. Best score for each metric and method in **bold**, second-best underlined.

| Method | Temp. | Accuracy (↑) | F1 (↑) | ECE (↓) | NLL (↓) | Brier (↓) |
|---|---|---|---|---|---|---|
| Single Network | 1.4 | **76.8** | **76.7** | 0.091 | 0.969 | 0.344 |
| Single Network | 1.6 | | | 0.061 | 0.928 | 0.334 |
| Single Network | 1.8 | | | 0.034 | **0.920** | **0.329** |
| Single Network | 2.0 | | | **0.029** | 0.939 | **0.329** |
| Single Network | 2.2 | | | 0.078 | 0.982 | 0.335 |
| Single Net w/ LoRA | 0.4 | **79.2** | **79.1** | 0.130 | 1.020 | 0.332 |
| Single Net w/ LoRA | 0.6 | | | 0.088 | 0.772 | 0.308 |
| Single Net w/ LoRA | 0.8 | | | 0.042 | 0.688 | 0.294 |
| Single Net w/ LoRA | 1.0 | | | **0.013** | **0.680** | **0.290** |
| Single Net w/ LoRA | 1.2 | | | 0.073 | 0.722 | 0.298 |
| MC Dropout | 0.4 | **76.6** | **76.6** | 0.203 | 1.554 | 0.372 |
| MC Dropout | 0.6 | | | 0.174 | 1.223 | 0.361 |
| MC Dropout | 0.8 | | | 0.111 | **1.114** | 0.344 |
| MC Dropout | 1.0 | | | **0.057** | 1.163 | **0.342** |
| MC Dropout | 1.2 | | | 0.175 | 1.333 | 0.393 |
| Explicit Ensemble | 1.0 | **79.8** | **79.9** | 0.100 | 0.744 | 0.285 |
| Explicit Ensemble | 1.2 | | | 0.072 | 0.719 | 0.282 |
| Explicit Ensemble | 1.4 | | | 0.041 | **0.718** | **0.281** |
| Explicit Ensemble | 1.6 | | | **0.019** | 0.737 | 0.284 |
| Explicit Ensemble | 1.8 | | | 0.046 | 0.777 | 0.290 |
| LoRA-Ensemble | 0.4 | **82.4** | **82.4** | 0.103 | 0.628 | 0.252 |
| LoRA-Ensemble | 0.6 | | | 0.063 | 0.565 | **0.247** |
| LoRA-Ensemble | 0.8 | | | **0.018** | **0.557** | **0.247** |
| LoRA-Ensemble | 1.0 | | | 0.034 | 0.587 | 0.253 |
| LoRA-Ensemble | 1.2 | | | 0.095 | 0.650 | 0.269 |

The forward pass through the backbone is parallelized by replicating the input along the batch dimension. In each LoRA module, the data is split into separate inputs per member and passed to the respective member with the help of a *vectorized map*, which allows a parallelized forward pass even through the LoRA modules. The outputs are then again stacked along the batch dimension. In this way, one makes efficient use of the parallelization on GPU, while at the same time avoiding loading the pre-trained backbone into memory multiple times. As a backbone for audio experiments, we use the Audio Spectrogram Transformer (AST) backbone Gong et al. (2021). That architecture was inspired by ViT (more specifically the data-efficient version of ViT akin to DeiT Touvron et al. (2020)) but is designed specifically for audio spectrograms. Following Gong et al. (2021), we initialize the audio model weights by transferring and appropriately interpolating them from ImageNet pre-training. See Appendix P and Q for details. As the AST version of LoRA-Ensemble would run into memory limits, we introduce chunking. While the forward pass through the backbone is still parallelized, the LoRA modules are called sequentially.[2]

Finally, the pre-trained model does not have the correct output dimension for our prediction tasks (i.e., it was trained for a different number of classes). Therefore we entirely discard its last layer and add a new one with the correct dimensions, which we train from scratch. Obviously, the weights of that last layer are different for every ensemble member. We parallelize it in the same way as the LoRA module described above.

---

[2]For the Explicit Ensemble the vectorization could not be used on GPU, due to a technical issue with the ViT implementation in PyTorch.

## M  TRAINING DETAILS OF LoRA-ENSEMBLE

The CIFAR-10/100, HAM10000, and iNaturalist 2017 dataset experiments are based on the ViT-Base-32 architecture Dosovitskiy et al. (2020). This model has 12 layers and uses 768-dimensional patch embeddings, and the multi-head attention modules have 12 heads. All Vision Transformer models for image classification are trained using the AdamW optimizer Loshchilov & Hutter (2017), except for INat2017, which is trained with SGD using a momentum of 0.9. The base learning rate is initially set to 0.0001 with a batch size of 32 for all experiments, except for INat2017, where a learning rate of 0.1 is used with a batch size of 128. Training employs a learning rate warm-up of 500 steps for all experiments, except for INat2017, which uses 2500 warm-up steps. During the warm-up phase, the learning rate increases linearly from 0 to the base value, after which it follows a cosine decay schedule for the remaining steps. For INat2017, an exponential learning rate decay with a factor of 0.94 is applied every 4 epochs. During the experiments, the gradients were calculated and then clipped not to exceed a maximum norm of 1. In the case of HAM10000, we used a weighted cross-entropy loss that considered the estimated effective number of samples, which was determined using a beta parameter of 0.9991 Cui et al. (2019). Uniform class weights were used for all other datasets. The maximum number of training epochs varies depending on the dataset. For CIFAR-10/100, the model is trained for 16 epochs (just over 25,000 steps), while for HAM10000 and INat2017, it is trained for 65 and 64 epochs, respectively. Overall, the hyperparameters used in this work were loosely based on Conrad (2023). The models were trained using pre-trained weights from `torchvision 0.17.1` on an NVIDIA Tesla A100 graphics card. Moreover, the LoRA models are configured with a rank of 8 for CIFAR-10/100, 4 for HAM10000, and 64 for INat2017. For Monte Carlo Dropout the dropout rate was empirically set to be 0.2. Refer to Appendix T for details.

The settings used for the ESC-50 dataset training are similar to those used in Gong et al. (2021). However, we used a batch size of 1 instead of 48 to enable training on a single GPU. The base learning rate is set to 0.00001 for the Explicit Ensemble as well as MC Dropout experiments and 0.00005 for LoRA-Ensemble. These learning rates are lower than the ones used in Gong et al. (2021), which is due to the smaller batch size. Refer to the Appendix R for more details. The LoRA models were implemented with a rank of 16. The dropout rate for MC dropout was kept at 0.2.

For language experiments on the SST-2 dataset Socher et al. (2013) we used the BERT base uncased model Devlin et al. (2019), loaded via the HuggingFace Transformers library Wolf et al. (2020). Training utilizes the AdamW optimizer Loshchilov & Hutter (2017) with $\beta_1 = 0.9$ and $\beta_2 = 0.999$, a linearly decaying learning rate over three epochs, and a batch size of 16. These settings were informed by prior work that used BERT on SST2 Chhablani (2023). We conduct a separate hyperparameter tuning for each method and select the learning rate from the candidate set $\{2 \times 10^{-6}, 7 \times 10^{-6}, 9 \times 10^{-6}, 2 \times 10^{-5}, 3 \times 10^{-5}, 5 \times 10^{-5}, 7 \times 10^{-5}\}$ that yields the highest accuracy. For MC Dropout, we used a dropout rate of 0.2. For all LoRA-based models we set the rank to 64 and use Xavier uniform initialization Glorot & Bengio (2010) for the LoRA layers, with a gain of 10.

As Fort et al. (2019a) have shown, varying initializations of the weights are most important to getting diverse ensemble members. For this reason, various initialization methods and corresponding parameters were tried, with a Xavier uniform initialization Glorot & Bengio (2010) with gain 10, giving the best combination of accuracy and calibration. For INat2017, a gain value of 1 is used. For more information, refer to Appendix N. This setting is kept for models across all datasets, including the one with an AST backbone.

For the same reason, we investigated whether adding noise to the pre-trained parameters of an Explicit Ensemble increases its performance through a higher diversity of members. However, the results did not show any additional benefits beyond what the randomly initialized last layer already provided, hence we did not use that option. For more details, refer to Appendix O.

## N  INITIALIZATION OF LoRA-ENSEMBLE PARAMETERS

Randomness in initialization is a key driver of diversity among ensemble members Fort et al. (2019a). Therefore, finding the right balance between diversity and overly disrupting parameters is crucial. Hu et al. (2021) propose using a random Gaussian initialization for $A$ while setting $B$ to

zero. This approach results in $\Delta W = BA$ being zero at the start of training. In our experiments, we adopt this pattern by always initializing $B$ to zero while varying the parameters and methods for initializing $A$. Following the method outlined by Hu et al. (2021), our initial experiments concentrated on the Gaussian initialization of $A$, with a mean $\mu = 0$ and varying standard deviations. Additionally, we tested the Xavier uniform initialization Glorot & Bengio (2010) using different values for the gain. All tests were conducted on the CIFAR-100 dataset and subsequently applied to other experiments. We compared results in terms of accuracy and Expected Calibration Error.

Table 18: Accuracy and Expected Calibration Error for different initialization methods and varying distribution parameters for LoRA-Ensemble.

| Init. Type | Std. / Gain | Accuracy ($\uparrow$) | ECE ($\downarrow$) |
|---|---|---|---|
| Gaussian | 0.02 | 81.2 | 0.041 |
| | 0.05 | 81.4 | 0.037 |
| | 0.1 | 81.7 | 0.035 |
| | 0.2 | 82.1 | 0.034 |
| | 0.5 | 82.6 | 0.036 |
| | 1 | 82.5 | 0.039 |
| | 2 | 81.7 | 0.046 |
| Xavier Uniform | 1 | 81.5 | 0.039 |
| | 5 | 82.2 | 0.034 |
| | 10 | 82.4 | 0.034 |
| | 15 | 82.6 | 0.037 |
| | 20 | 82.4 | 0.038 |
| | 30 | 82.2 | 0.043 |

In Tab. 18, the results are quantitatively presented. It is immediately evident that both techniques and all tested parameters perform similarly. While more specialized models may surpass our results in terms of accuracy, our primary focus is on calibration, with the goal of maintaining comparable predictive performance. Visual inspection of the results in Fig. 16 confirms the high similarity among all results. Choosing a small calibration error while maintaining high accuracy as a decision criterion, both Gaussian initialization with a standard deviation of 0.5 and Xavier uniform initialization with a gain of 10 or 15 are viable candidates. Since a gain of 10 combines high accuracy with the lowest Expected Calibration Error, we select Xavier uniform initialization with a gain of 10 for our experiments.

## O  INITIALIZATION OF EXPLICIT ENSEMBLE PARAMETERS

A pre-trained Vision Transformer model is the backbone for our computer vision experiments. Correspondingly, the parameters of all members in an Explicit Ensemble are initialized to the same values across members. Initialization is a primary driver of diversity in ensemble members Fort et al. (2019a). Hence, it is crucial to study the effect of noise in the parameter initialization on the calibration of the resulting ensemble. In the case of pre-trained model weights not having been trained on a dataset with the same number of classes, the last layer of all models is replaced completely. This means that regardless of the ensemble technique used, the weights of the last layer, which is responsible for classification, will vary across members. This variation in the weights of the classification layer is expected to contribute significantly to the diversity of the members. Nonetheless, we studied the impact of adding noise to the parameters of an Explicit Ensemble. This was done using the following formula:

$$W_{\text{new}} = W + \alpha \cdot dW, \tag{14}$$

where $dW \sim \mathcal{N}(0, \sigma_W)$. Here $\alpha$ is a scale factor to control the amount of noise and $\sigma_W$ is the standard deviation of the parameters within a weight matrix. This was applied to all weight matrices separately.

It is expected that the initial layers of a neural network will learn basic features, while the later layers will include dataset-specific properties. Therefore, it is assumed that adding noise to the later layers would increase diversity while maintaining pre-training. However, adding noise to the earlier layers

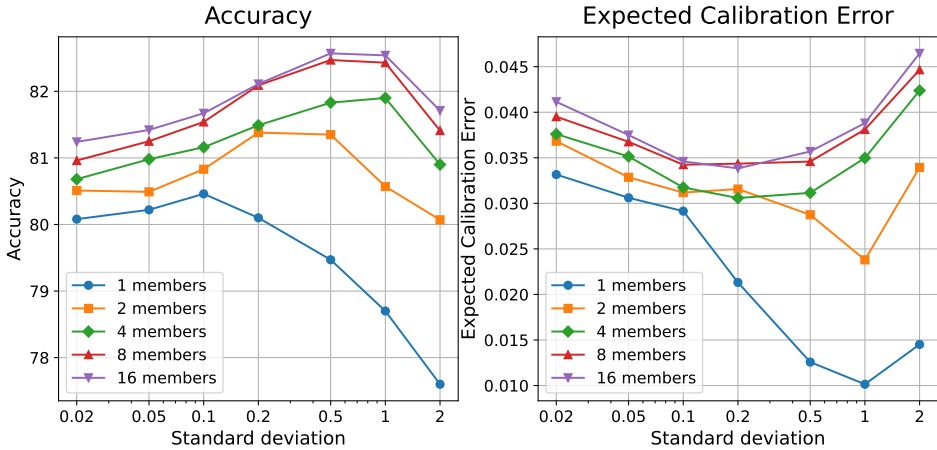

(a) Gaussian initialization with varying standard deviation.

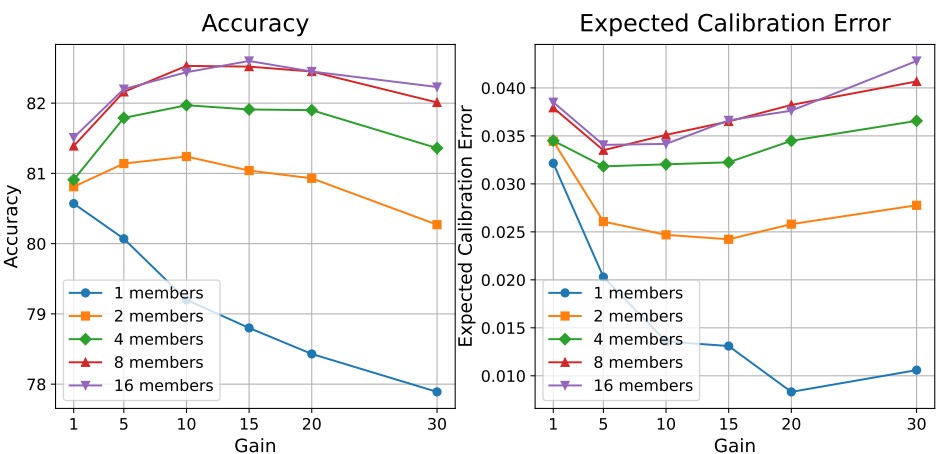

(b) Xavier uniform initialization with varying gain

Figure 16: Accuracy and Expected Calibration Error for different initialization methods and varying distribution parameters across different ensemble sizes for LoRA-Ensemble.

might disrupt pre-training more significantly, especially with smaller datasets, as these parameters may not converge to meaningful values again. To address this, an experiment was set up where noise was added only to the last encoder layers of the model, increasing the number of affected encoder layers gradually. Additionally, several different noise scales $\alpha$ were tried, ranging from 1 to 0.0001. In the presented experiment, the last classification layer is initialized using PyTorch's default method for linear layers. At the time of writing it is as follows:

$$W_{\text{init}} = \text{Unif}\left(-\sqrt{5} \cdot \sqrt{\frac{3}{fan\_in}}, \sqrt{5} \cdot \sqrt{\frac{3}{fan\_in}}\right) \tag{15}$$

$$B_{\text{init}} = \text{Unif}\left(-\sqrt{\frac{1}{fan\_in}}, \sqrt{\frac{1}{fan\_in}}\right). \tag{16}$$

Here $W$ specifies the weight matrix and $B$ is the bias. Experiments are conducted on the CIFAR-100 dataset.

## O.1 RESULTS

The most important metrics for this section are accuracy and Expected Calibration Error. The results for adding noise to the last layer up to the last five layers are summarized in Fig. 17. Fig. 17a depicts the results for a single model, while Fig. 17b shows the results for an ensemble of 16 members.

It is evident that none of the experiments surpass the baseline of not using any additional noise beyond the random initialization of the last classification layer. After the last five layers, the results become uninteresting, as they do not vary significantly from those shown in the plots. Therefore, the presentation is truncated at five layers. Based on the presented results, no additional noise is injected into the Explicit Ensemble, and only the last layer initialization is varied.

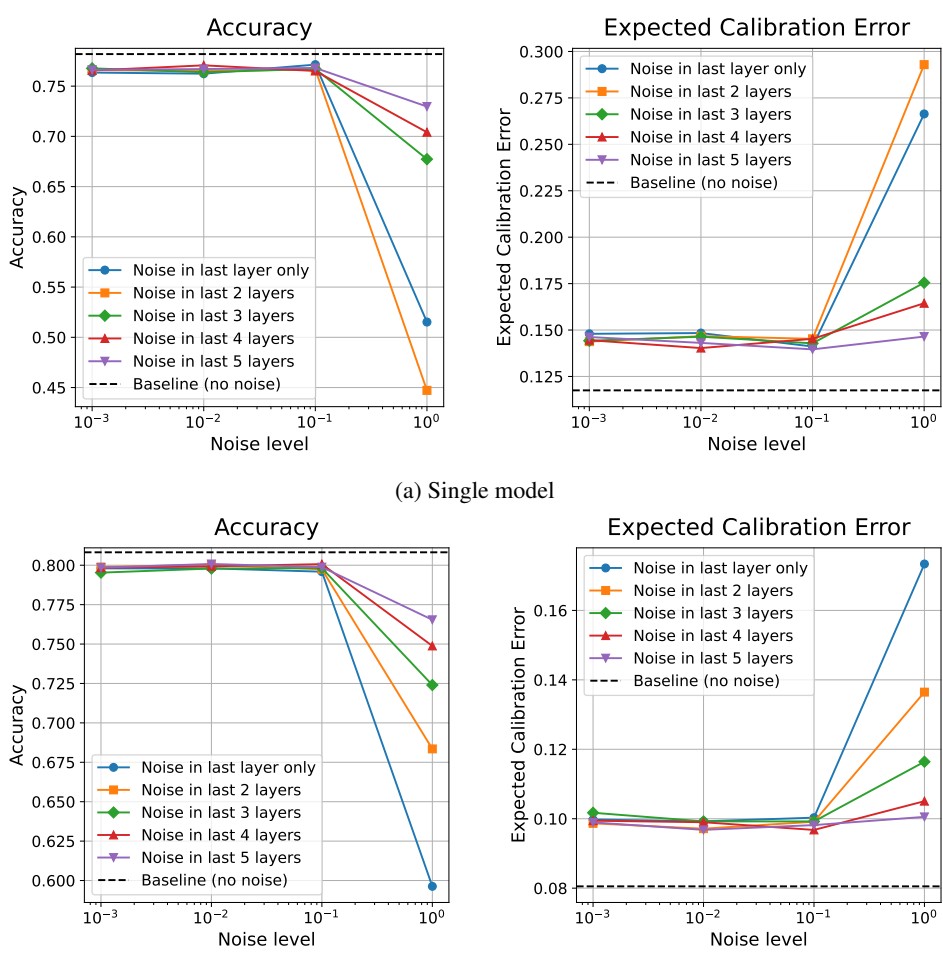

(a) Single model

(b) 16 ensemble members

Figure 17: Accuracy and Expected Calibration Error for different noise levels across varying numbers of layers for the Explicit Ensemble. The baseline with no noise is indicated by a dashed black line.

## P  AST IMPLEMENTATION

A different backbone is used for the experiment on the audio dataset. Specifically, we use the Audio Spectrogram Transformer (AST) following the implementation of Gong et al. (2021), with slight modifications to fit our general architecture. Appendix Q demonstrates the equivalence of our implementation. In their experiments, Gong et al. (2021) used two different types of pre-trained weights: one pre-trained on a large image dataset and the other on an audio dataset. For our research, we transfer the weights of a vision transformer model known as DeiT Touvron et al. (2020), which has been pre-trained on the ImageNet dataset Deng et al. (2009), to the original AST architecture by Gong et al. (2021). The model has 12 layers, uses 768-dimensional patch embeddings, and the multi-head attention modules have 12 heads. This task is considered more challenging than using models pre-trained on audio datasets.

## Q  VALIDATION OF AST IMPLEMENTATION

The Audio Spectrogram Transformer (AST) model provided by Gong et al. (2021) was copied without any changes. However, the training and evaluation pipeline was adapted to fit our architecture. Correspondingly, it was essential to validate the equivalence of our implementation by training a single AST on the ESC-50 dataset. The results of our model should closely match those provided in Gong et al. (2021).

They offer two sets of pre-trained weights: one where the weights of a Vision Transformer pre-trained on ImageNet Deng et al. (2009) are transferred to AST, and another where the AST was pre-trained on AudioSet Gemmeke et al. (2017). To verify our implementation, we ran it using the settings provided by Gong et al. (2021) and compared the results, which are summarized in Tab. 19. The results for both pre-training modes fall within the uncertainty range provided by Gong et al. (2021). This suggests that our pipeline yields comparable outcomes, validating our implementation for continued use.

Table 19: Comparison of the results obtained for the AST as given in Gong et al. (2021) and those obtained by our implementation. AST-S refers to the AST pre-trained on ImageNet, and AST-P refers to the AudioSet pre-training. Both results fall within the uncertainty range provided by Gong et al. (2021).

| Model | Accuracy (Gong et al., 2021) | Accuracy (our implementation) |
|-------|------------------------------|-------------------------------|
| AST-S | $88.7 \pm 0.7$ | 88.0 |
| AST-P | $95.6 \pm 0.4$ | 95.8 |

## R  HYPER-PARAMETER TUNING FOR AST EXPERIMENT

The original training settings of the AST-S model in Gong et al. (2021) utilize a batch size of 48. However, due to the memory constraint of single GPU training on an NVIDIA Tesla A100 with 80 GB memory, replicating a batch size of 48 as in the original publication was infeasible for training an Explicit AST-S Ensemble with 8 members. Consequently, we perform minimal hyper-parameter tuning by employing a batch size of 1 for both the explicit AST-S and the LoRA AST-S model, exploring various learning rates. Apart from batch size and learning rate adjustments, all other settings remain consistent with Gong et al. (2021).

The hyper-parameter tuning results for the explicit model using a batch size of 1, as shown in Tab. 20, demonstrate performance similar to the original implementation with a batch size of 48, allowing for a fair comparison with our method Gong et al. (2021). Additionally, Tab. 21 showcases the outcomes of tuning the learning rate for our LoRA AST-S model.

Table 20: Single model 5-Fold cross-validation results of AST-S on ESC-50 sound dataset with different learning rates and batch size 1. The model settings selected based on accuracy for the experiments are **highlighted**.

| Model | Learning rate | Accuracy ($\uparrow$) | ECE ($\downarrow$) |
|-------|---------------|----------------------|--------------------|
| **AST-S** | **0.00001** | **88.2** | **0.0553** |
| AST-S | 0.00005 | 81.7 | 0.0933 |

## S  COMPUTATIONAL COST FOR AST MODELS

Similarly to the way we did for the Vision Transformer models, we estimate the required resources for AST models. The resource needs are presented in Tab. 22. The number of parameters is reported for an ensemble of 8 members, with the $A$ and $B$ matrices in models using LoRA having a rank of 16. Training and inference times were measured on a single NVIDIA Tesla A100-80GB GPU, with a batch size of 1. Training time is given as the average wall clock time per training epoch while

Table 21: Single model 5-Fold cross-validation results for our LoRA AST-S implementation on ESC-50 sound dataset with different learning rates and batch size 1. The model settings selected based on accuracy for the experiments are **highlighted**.

| Model | Learning rate | Accuracy ($\uparrow$) | ECE ($\downarrow$) |
|---|---|---|---|
| LoRA AST-S | 0.00001 | 85.6 | 0.0447 |
| **LoRA AST-S** | **0.00005** | **87.9** | **0.0487** |
| LoRA AST-S | 0.0001 | 84.7 | 0.0501 |
| LoRA AST-S | 0.0005 | 24.1 | 0.0291 |
| LoRA AST-S | 0.001 | 11.8 | 0.0295 |

Table 22: Parameter counts and computation times for an Explicit Ensemble of 8 AST models and the corresponding LoRA-Ensemble. Training time is the average duration for one epoch on ESC-50, with batch size 1. Inference time is the average duration of a forward pass, with batch size 1.

| Method | Parameter overhead | Training time [s] | Inference time [ms] |
|---|---|---|---|
| Explicit Ensemble | $8 \times 87\text{M}$ | 517 | $8 \times 7.3$ |
| LoRA-Ensemble | $1.08 \times 87\text{M}$ | 348 | 73.9 |

training on ESC-50, with 8 ensemble members. Inference time is reported as the average time for a single forward pass of an ESC-50 sample with a batch size of 1.

As mentioned in Appendix L, the Explicit Ensemble processes the members sequentially, while LoRA-Ensemble is parallelized. However, fully parallelizing the training of AST models causes memory issues, so chunking was introduced. Thus, in LoRA-Ensemble models, the pass through the backbone runs in parallel, while LoRA modules are called sequentially. This also explains the significantly higher inference time compared to the results in Sec. A.9. Additionally, the one-time delay incurred by PyTorch's *vmap* function causes LoRA-Ensemble to be slightly slower at inference time.

## T   HYPERPARAMETER TUNING FOR MC DROPOUT

We conducted an analysis to determine the impact of dropout probability on the accuracy and calibration of the ViT with Monte Carlo dropout. Fig. 18 displays the accuracy and ECE scores for various dropout probabilities. The experiment is carried out on the HAM10000 dataset with 16 members. Our findings show that a dropout probability of 0.2 offers a good balance between accuracy and calibration.

## U   SNAPSHOT ENSEMBLE IMPLEMENTATION DETAILS

Snapshot Ensemble Huang et al. (2017), in its pure form, consists of training a single model with cycling learning and taking snapshots every few epochs. This can make it hard, however, for the model to converge to anything meaningful within the low number of epochs available for training per snapshot. Therefore, Snapshot Ensemble was modified slightly, by first letting training run for a number of epochs, without any cycling of the learning rate. After this burn-in period the learning rate is at 0 and a first snapshot is taken. The remaining number of epochs is split evenly. If the remaining number of epochs is not divisible by the desired number of ensemble members, the burn-in period is extended until it is. For the HAM10000 dataset training is left at 65 epochs, with 20 burn-in epochs. For CIFAR-10 and CIFAR-100 using only 16 epochs would only leave 1 epoch per cycle for bigger models. Therefore, training is extended to 30 epochs with a burn-in period of 15 epochs.

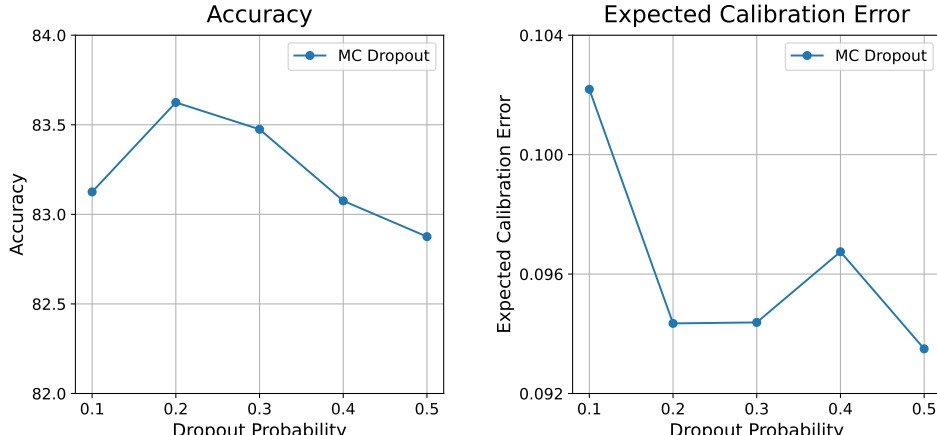

Figure 18: Accuracy and Expected Calibration Error for different dropout probabilities methods for MC Dropout on HAM10000 dataset.

## V    IMPLICIT ENSEMBLE BASELINE CHALLENGE

Many implicit ensemble methods, such as those proposed in Wen et al. (2020); Turkoglu et al. (2022); Durasov et al. (2020); Havasi et al. (2020), are architecture-specific and predominantly designed for MLPs or CNNs. As a result, adapting these techniques to transformer architectures presents significant challenges, since transformers' computation structure is quite different than MLPs and CNNs.

In particular, we attempted to implement FiLM-Ensemble Turkoglu et al. (2022) on a self-attention network, given the promising results reported by its authors. However, the authors themselves noted that applying FiLM-Ensemble to transformers is not straightforward, mainly because transformers rely on LayerNorm, whereas FiLM-Ensemble was developed with BatchNorm in mind. Our experiments confirmed that directly using BatchNorm in transformers led to notable performance degradation. We explored several approaches to adapt LayerNorm, but the most effective results were achieved by fixing all affine parameters for each ensemble member. This allowed for slight initial variations to introduce randomness and diversity, while keeping the variation among members minimal. The results, summarized in Tab. 23, show that increasing the ensemble size slightly improved accuracy, though the Expected Calibration Error (ECE) fluctuated without consistent improvement. In fact, when using larger ensemble sizes, such as 8 or 16, both accuracy and calibration worsened across all settings we tested.

Table 23: Performance of FiLM-Ensemble for Vision Transformer (ViT) on CIFAR-10. Increasing the ensemble size slightly improves accuracy, but ECE fluctuates without showing consistent improvement.

| # ensemble members | Accuracy (↑) | ECE (↓) |
| --- | --- | --- |
| 1 | 90.54 | 0.0286 |
| 2 | 91.18 | **0.0269** |
| 4 | **91.23** | 0.0289 |

## W    BAYESIAN LORA

Bayes-LoRA Yang et al. (2024) introduces a Bayesian approach on the LoRA adapter parameters by fitting a Gaussian posterior around the maximum a posteriori (MAP) estimate of the fine-tuned model. In practice, this means we first obtain a standard LoRA fine-tuned network and then apply a Laplace approximation over its adapter weights. To make this tractable at scale, Bayes-LoRA relies on a Kronecker-factored approximation of the Hessian, which allows efficient estimation of the

posterior covariance. The result is a Bayesian model that can capture uncertainty while remaining computationally efficient compared to traditional Bayesian neural networks.

We evaluate Bayes-LoRA on the SST-2 sentiment classification task using a BERT base uncased Socher et al. (2013); Devlin et al. (2019) backbone. The method is applied in a post-hoc fashion after fine-tuning. The original implementation of Yang et al. (2024) used[3]. For MAP training, we follow the standard LoRA setup with a learning rate of $5 \times 10^{-5}$, training for 3 epochs with batch size 16. The LoRA rank is set to 64, identical to the LoRA-Ensemble. The prior variance is chosen as $10^{-3}$, since larger values tend to degrade performance. To balance computational efficiency and uncertainty estimation, the number of posterior samples (i.e., ensemble members) is fixed at 512. For the Kronecker-factored approximation, we use $n_{\text{kfac}} = 10$.

In terms of results, Bayes-LoRA falls short of LoRA-Ensemble and Explicit Ensemble methods in predictive performance, measured by accuracy and F1. This observation is in line with previous findings in the literature, such as Daxberger et al. (2021). We attribute this limitation to the reliance on a local Gaussian approximation around a single MAP solution, in contrast to the diversity gained through sufficiently independent ensemble members. However, the main strength of Bayes-LoRA lies in its ability to capture predictive uncertainty effectively, reaching a level comparable to both LoRA-Ensemble and Explicit Ensemble. Detailed results can be found in Tab. 8 in Appendix A.7.

From an efficiency perspective, Bayes-LoRA requires significantly more computation at inference: evaluating a single test example takes roughly 250 ms (512 posterior samples are used), compared to 22.7 ms (Tab. 9) for LoRA-Ensemble (16 members are used). This overhead makes Bayes-LoRA impractical for real-time applications but potentially valuable in settings where predictive uncertainty is crucial and strict latency constraints are less relevant.

## X    DEFINITIONS OF EVALUATION METRICS

We primarily evaluate our models on accuracy and Expected Calibration Error (ECE, Guo et al., 2017). In addition to accuracy and Expected Calibration Error, we have calculated several other scores that have been used in the context of probabilistic deep learning. In the following section, we present the formulations used in our implementations.

### X.1    ACCURACY

The accuracy is implemented instance-wise as follows:

$$\text{Acc} = \frac{1}{N} \sum_{i=1}^{N} \frac{|\hat{y}_i \cap y_i|}{|\hat{y}_i \cup y_i|} \tag{17}$$

Here $y_i$ denotes the true label of the sample $i$, $\hat{y}_i$ is the predicted label of the sample $i$, and $N$ means the total number of samples.

### X.2    EXPECTED CALIBRATION ERROR

The Expected Calibration Error is a widely used metric for measuring the calibration of neural networks. We use the definition given in Guo et al. (2017). ECE is defined as the expected difference between accuracy and confidence across several bins. We first need to define accuracy and confidence per bin $B_m$ as follows:

$$\text{Acc}(B_m) = \frac{1}{|B_m|} \sum_{i \in B_m} \mathbf{1}(\hat{y}_i = y_i), \tag{18}$$

$$\text{Conf}(B_m) = \frac{1}{|B_m|} \sum_{i \in B_m} \hat{p}_i. \tag{19}$$

---

[3] https://github.com/MaximeRobeyns/bayesian_lora

Again, $y_i$ and $\hat{y}_i$ denote the true and predicted labels of sample $i$ respectively, and $\hat{p}_i$ is the predicted confidence of sample $i$. With this the Expected Calibration Error is given as:

$$\text{ECE} = \sum_{m=1}^{M} \frac{|B_m|}{n} |\text{Acc}(B_m) - \text{Conf}(B_m)| \tag{20}$$

### X.3 MACRO F1-SCORE

$$F1 = \frac{1}{C} \sum_{j=1}^{C} \frac{2p_j r_j}{p_j + r_j}, \tag{21}$$

where $r_j$ represents the Recall of class $j$, defined as $r_j = \frac{TP}{TP+FN}$, and $p_j$ represents the Precision of class $j$, defined as $p_j = \frac{TP}{TP+FP}$, and $C$ refers to the number of classes, Here, $TP$, $FP$, and $FN$ denote True Positives, False Positives, and False Negatives respectively.

### X.4 NEGATIVE LOG-LIKELIHOOD (NLL)

$$NLL = -\frac{1}{N} \sum_{i=1}^{N} \sum_{j=1}^{C} (y_{i,j} \log \hat{p}_{i,j}) = -\frac{1}{N} \sum_{i=1}^{N} \log \hat{p}_i, \tag{22}$$

where $N$ denotes the number of datapoints, $C$ the number of classes, $y_{i,j}$ is 1 if the true label of point $i$ is $j$ and 0 otherwise and $\hat{p}_{i,j}$ is the predicted probability of sample $i$ belonging to class $j$.

### X.5 BRIER SCORE

For Brier score we take the definition by Brier (1950), which is as follows:

$$BS = \frac{1}{N} \sum_{i=1}^{N} \sum_{j=1}^{C} (\hat{p}_{i,j} - y_{i,j})^2, \tag{23}$$

where $N$ denotes the number of datapoints, $C$ the number of classes, $y_{i,j}$ is 1 if the true label of point $i$ is $j$ and zero otherwise and $\hat{p}_{i,j}$ is the predicted probability of sample $i$ belonging to class $j$.

### X.6 AREA UNDER THE RECEIVER OPERATING CHARACTERISTIC CURVE (AUROC)

The AUROC score evaluates the performance of a binary classifier by measuring its ability to distinguish between positive and negative classes, as introduced by Hanley & McNeil (1982). In our out-of-distribution (OOD) detection experiments, the positive class corresponds to an in-distribution sample, while the negative class corresponds to an out-of-distribution sample.

The AUROC is computed as the area under the ROC curve, which plots the true positive rate (TPR) against the false positive rate (FPR) across various decision thresholds. The TPR and FPR are defined as follows:

$$\text{TPR} = \frac{\text{TP}}{\text{TP} + \text{FN}}, \tag{24}$$

$$\text{FPR} = \frac{\text{FP}}{\text{FP} + \text{TN}}, \tag{25}$$

where TP, FP, FN, and TN represent the true positives, false positives, false negatives, and true negatives, respectively.

The AUROC score is given by the following integral:

$$\text{AUROC} = \int_0^1 \text{TPR(FPR)}, d\text{FPR}. \tag{26}$$

A higher AUROC score indicates better classification performance, with a score of 1 representing a perfect classifier, and a score of 0.5 indicating performance equivalent to random chance.

### X.7 AREA UNDER THE PRECISION-RECALL CURVE (AUPRC)

The Area Under the Precision-Recall Curve (AUPRC) assesses the performance of a binary classifier by measuring its ability to accurately identify positive instances, as described by Davis & Goadrich (2006). In our out-of-distribution (OOD) detection experiments, the positive class corresponds to in-distribution samples, while the negative class corresponds to out-of-distribution samples.

The AUPRC is calculated as the area under the Precision-Recall (PR) curve, which plots precision against recall at various decision thresholds. Precision and recall are defined as follows:

$$\text{Precision} = \frac{\text{TP}}{\text{TP} + \text{FP}}, \tag{27}$$

$$\text{Recall} = \frac{\text{TP}}{\text{TP} + \text{FN}}, \tag{28}$$

where TP, FP, and FN represent true positives, false positives, and false negatives, respectively.

The AUPRC score is the integral of precision with respect to recall, expressed as:

$$\text{AUPRC} = \int_0^1 \text{Precision(Recall)} \, d\text{Recall}. \tag{29}$$

A higher AUPRC score indicates better classifier performance in recognizing positive instances, with a score near 1 representing a good classifier, characterized by both high recall and high precision. This metric is especially valuable for evaluating classifiers on imbalanced datasets.

## Y STATEMENT ON THE USE OF GENERATIVE AI AND DECLARATION OF ORIGINALITY

In the preparation of this thesis, generative AI (ChatGPT version 4o) was utilized for language corrections, including grammar and style improvements. The use of AI was limited to improving readability; it was not used to generate original content, conduct research, or contribute to the intellectual development of the work.

