# OpenReview forum: "LoRA-Ensemble: Efficient Uncertainty Modelling for Self-Attention Networks"
_ICLR.cc/2026/Conference — ICLR 2026 Conference Withdrawn Submission_

### Official Review · Reviewer_eNJu · 2025-10-28

**Soundness:** 2
**Presentation:** 3
**Contribution:** 1
**Rating:** 2
**Confidence:** 4

**Summary:**

The paper proposes an implicit ensemble for Transformers in which each member has its own LoRA adapters on the attention projections. Predictions from the members are averaged. The authors argue that diversity is high in both function and weight space, and they report superior performance over competing methods across several tasks and datasets.

**Strengths:**

* The paper is well written.
* The idea of LoRA-Ensemble is straightforward to use.

**Weaknesses:**

* The novelty of their method is limited. LoRA-Ensemble is very close to prior weight-sharing ensembles (esp. BatchEnsemble). The paper’s main distinction is additive low-rank updates vs. BatchEnsemble’s multiplicative rank-1 modulation. While useful, this reads incremental and largely engineering-driven; there is no principled account that predicts when/why additive low-rank adapters should dominate multiplicative modulations for accuracy/calibration.
* The experiments in the paper focus on small-scale datasets and architectures. There is no evaluation on ImageNet-O or ImageNet-C. The paper would be considerably stronger with experiments on these datasets and ablations at ViT-B/L scales. particularly relevant given LoRA’s typical use with larger models.
* The manuscript leans on descriptive analyses (function-space spread, SVD “intruder dimensions”) and a qualitative claim of unique learning dynamics, but offers no formal framework linking these observations to improved calibration or diversity beyond visualization.

Minor weakness:

* The results for CIFAR100 for the Batch ensemble are different from the original paper, and also, the superior performance of the Explicit Ensemble across different metrics is reported in multiple previous works (E.g., SNGP, DUQ, Batch ensemble, Mask ensemble) on CIFAR 100/10 Benchmark, but, surprisingly, it has lower performance compared to MC-dropout in the papers’ experiments. I think the author should spend more time on hyperparameters and experiment setup for the competitors.

**Questions:**

* Why does Single Net w/ LoRA outperform the Explicit ensemble, especially from a diversity point of view?

---

### Official Review · Reviewer_LNJm · 2025-10-31

**Soundness:** 4
**Presentation:** 3
**Contribution:** 3
**Rating:** 6
**Confidence:** 3

**Summary:**

The paper presents a lora-driven ensemble to generate uncertainty estimates for a large LLM model.

**Strengths:**

well done study and a well written paper

The experimental results are actually pretty detailed and well developed.

**Weaknesses:**

- Use of an ensemble to generate uncertainty estimate seem to be counter intuitive to the idea of why LORA is used. That is, to reduce the number of parameters that need to be trained. Replacing this with an ensemble just seems to spoil that idea. That being said, it is still going to be better than creating an ensemble with the original parameter space.


- The insights into what and why LORA based uncertainty estimates is useful and why it provides notions of uncertainty is actually hidden in section 4. I think, this section on enhanced diversity is the way to address my question on first weakness. I recommend the authors to highlight this in the main parts of the paper, and in fact, base their contribution on this. I would like some way of explaining the need for LORA ensembles on this section saying, enhanced diversity in population is observed due to the presence to low rank factorization. This is a very good argument to make and would strengthen the paper.

- I would also make the training trajectories plot a little bit more readable.

**Questions:**

The key question that substantiate contribution is, is an ensemble way of generating uncertainty estimates actually useful/practical for such large models. At some point, I would like this question addressed rather than just doing an ensemble because ensemble-based uncertainty estimates are the most popular/easy way of doing things

I would also make the training trajectories plot a little bit more readable.

Are you initializing the A and B matrices both through some random initialization or have something fixed. I am thinking, fixing one of matrices using the basis space of the weight matrix would provide the notion of transformability between the low rank and the actual space. This provides a way to transferring uncertainty between the low rank space and the actual full rank space.

---

### Official Review · Reviewer_3TYo · 2025-11-01

**Soundness:** 3
**Presentation:** 2
**Contribution:** 2
**Rating:** 4
**Confidence:** 3

**Summary:**

This paper introduces LoRA-Ensemble, which is a new, efficient ensemble method where the individual ensemble members are trained based on a single pre-trained self-attention model using Low-Rank Adaptation (LoRA). They apply this ensembling method on a range of classifications tasks and compare it to both implicit (e.g., dropout) as well as explicit ensembling methods. They find that LoRA-Ensemble match or exceed the accuracy of the other methods and improve on the other methods in terms of calibration across tasks. Lastly, they analyze the diversity of ensemble members in LoRA-Ensemble and in explicit ensembles and find that LoRA-Ensemble exhibit a higher diversity.

**Strengths:**

The paper uses the simple (and therefore potentially very widely applicable) idea to create ensemble members by fine-tuning a single pre-trained transformer model using LoRA and comprehensively tests it on a wide range of relevant datasets against a number of baselines. The experimental results are described in detail and look promising. Therefore, the proposed method could provide an effective, yet much more compute-efficient way of training ensembles of self-attention networks.
Furthermore, the relevant background on LoRA is explained clearly and concisely and the paper provides a comprehensive overview over related work.

**Weaknesses:**

1. The paper could partially profit from more clarity in the writing/presentation of the results. Concretely, for example, the first part of the 'experiments' section switched around from paragraph to paragraph between explaining the datasets used, the baselines evaluated against and the metrics used. Similarly, I felt like the explanation in the section on 'Enhanced Diversity In LoRA-Ensemble' was often switching around between various methods without having a clear storyline, which made it hard to evaluate. Furthermore, Figure 2 and Figure 3 were missing proper captions.
2. The main paper was missing details on the implementation of the LoRA-Ensemble. Concretely, in the experiments section, I found it hard to understand which pre-trained base model was used for which task, how they performed the LoRA-finetuning and how they ensured that comparison to the baselines was fair.

**Questions:**

1. Where can I find this contribution in the main paper; or is this only contained in the appendix (then it might be useful to make this clear):
> We conduct extensive empirical analyses of how LoRA rank, initialization scheme, model scale, and parameter-sharing strategies impact performance, and we adapt LoRA-Ensemble for convolutional neural networks (CNNs) to demonstrate its broad applicability.
2. As written in the 'weaknesses', I was missing details on the implementation of the LoRA-Ensemble. Could you expand on this aspect?
3. In line 170-171, what do you mean by 'empirically set'? Do you mean that you used hyperparameter optimization for choosing this parameter?
4. I found the following point quite interesting (it came up multiple times in the paper). How do you explain this? Can you give more detail on how you trained the ensembles members of the explicit ensemble? Were they also fine-tuned base on a pre-trained model? Here is the point I am referring to:
> apparently a consequence of the fact that already a single ViT model, and thus every ensemble member, benefits from the addition of LoRA.
5. Why are there less baselines that you are comparing to in Table 3?
6. Could you extend on the individual methods used in the analysis of the enhanced diversity in LoRA-Ensemble. For example, why do you believe that that using t-SNE to visualize the spread in function space (or the development over the training trajectories as in Figure 3) is giving relevant evidence for the ensemble members having higher diversity?

---

### Official Review · Reviewer_oHiP · 2025-11-01

**Soundness:** 3
**Presentation:** 3
**Contribution:** 2
**Rating:** 4
**Confidence:** 3

**Summary:**

The paper proposes LoRA-Ensemble, a parameter-efficient ensemble method for self-attention networks.
Instead of maintaining multiple full copies of a transformer, the approach freezes the shared backbone and attaches distinct LoRA modules to create ensemble members. Each member’s low-rank matrices modulate the attention projections, producing diversity with negligible memory overhead.
Across benchmarks, the method claims to achieve accuracy and calibration on par with—or superior to—explicit ensembles.

**Strengths:**

1. Strong Empirical Coverage and Rigor: The authors evaluate on large-scale and cross-modal datasets (iNaturalist, ESC-50, SST-2) with clear error bars, runtime/memory analysis, and calibration metrics (ECE, NLL, Brier).
2. Clear and Practical Contribution – The paper demonstrates that LoRA modules can serve as lightweight ensemble “handles” for transformers. The design is simple, easy to integrate, and compatible with existing pre-trained models.
3. Clarity and Presentation – The manuscript is well-organized, easily understandable, and visually clear; figures and appendices make the results reproducible.

**Weaknesses:**

1. Limited Theoretical Foundation – The work still lacks a rigorous theoretical analysis of why LoRA-based low-rank perturbations yield well-calibrated ensembles or maintain diversity. The paper remains empirical in nature. There are many ways to create an "ensemble of models"  from a template one but it is unclear why multiple LoRA models create an useful ensemble.

2. Conceptual Overlap with BatchEnsemble – Although the authors includes BatchEnsemble comparisons and explanations, the underlying idea (shared backbone + per-member low-rank factors) remains very close. The distinction is mainly architectural (placement in self-attention) rather than conceptual.

3. Computational Complexity per Batch – Despite parameter efficiency, inference remains O(N) in ensemble size since each member requires a forward pass. The paper acknowledges this but does not propose a remedy.

4. Lack of Broader Theoretical Framing – The method could benefit from a probabilistic interpretation (e.g., posterior sampling view) or connection to Bayesian low-rank inference.

5. Minor Breadth Gaps – While CNN and text extensions are shown, results for very large LLMs or more challenging ImageNet-scale ViT variants would strengthen generality claims.

**Questions:**

What underlying principle explains LoRA-Ensemble’s improved calibration compared to both explicit and implicit baselines?

Can the authors formalize or intuitively explain how LoRA-based perturbations produce higher functional diversity than other methods? Currently, this is shown only via SVD and cosine similarity plots. The argument for “enhanced diversity” is entirely empirical.

---

### Note · Authors · 2025-11-20

I have read and agree with the venue's withdrawal policy on behalf of myself and my co-authors.